RESEARCH COMMUNICATION

# Defining host–pathogen interactions employing an artificial intelligence workflow

Daniel Fisch[1†], Artur Yakimovich[2†], Barbara Clough[1], Joseph Wright[1], Monique Bunyan[1], Michael Howell[3], Jason Mercer[2], Eva Frickel[1*]

[1]Host-Toxoplasma Interaction Laboratory, The Francis Crick Institute, London, United Kingdom; [2]MRC-Laboratory for Molecular Cell Biology, University College London, London, United Kingdom; [3]HTS, The Francis Crick Institute, London, United Kingdom

**Abstract** For image-based infection biology, accurate unbiased quantification of host–pathogen interactions is essential, yet often performed manually or using limited enumeration employing simple image analysis algorithms based on image segmentation. Host protein recruitment to pathogens is often refractory to accurate automated assessment due to its heterogeneous nature. An intuitive intelligent image analysis program to assess host protein recruitment within general cellular pathogen defense is lacking. We present HRMAn (Host Response to Microbe Analysis), an open-source image analysis platform based on machine learning algorithms and deep learning. We show that HRMAn has the capacity to learn phenotypes from the data, without relying on researcher-based assumptions. Using *Toxoplasma gondii* and *Salmonella enterica* Typhimurium we demonstrate HRMAn's capacity to recognize, classify and quantify pathogen killing, replication and cellular defense responses. HRMAn thus presents the only intelligent solution operating at human capacity suitable for both single image and high content image analysis.
**Editorial note:** This article has been through an editorial process in which the authors decide how to respond to the issues raised during peer review. The Reviewing Editor's assessment is that all the issues have been addressed (see decision letter).
DOI: https://doi.org/10.7554/eLife.40560.001

*For correspondence:
eva.frickel@crick.ac.uk

†These authors contributed equally to this work

Competing interests: The authors declare that no competing interests exist.

## Introduction

High content imaging (HCI) has revolutionized the field of host–pathogen interaction by allowing researchers to perform image-based large-scale compound and host genome-wide depletion screens in a high-throughput fashion (*Brodin and Christophe, 2011*; *Mattiazzi Usaj et al., 2016*). The majority of these screens assess host–pathogen interactions using bulk colorimetric or automated enumeration of pathogen growth at the population level (*Ang and Pethe, 2016*; *Radke et al., 2018*). Additionally, quantification of host–pathogen interaction (e.g. analysis of host protein recruitment to the pathogen) in general is often performed manually. However, to meaningfully dissect cell-mediated pathogen control, it is imperative to quantify the host response and pathogen fate at the single-cell level. Many open-source, e.g. CellProfiler (*Carpenter et al., 2006*), and proprietary, e.g. Perkin Elmer Harmony, analysis software packages have been developed and successfully employed for this purpose (*Eliceiri et al., 2012*; *Stöter et al., 2013*; *Smith et al., 2018*). To advance the state of the art in image analysis of host–pathogen interaction, incorporation of cutting-edge machine intelligence algorithms (*Simonyan and Zisserman, 2014*; *LeCun et al., 2015*) to stratify the image content without the requirement to program complex integrations is needed. HRMAn relies on the same well-established image segmentation strategies as many other programs but

offers an intuitive integration of deep learning for more complex image analysis. Solutions existing to date can be split into two major categories: user-friendly turn-key GUI (TK-GUI)-based solutions and scripts ensembles (SE) solutions. Due to the large support burden of the TK-GUI, these programs lack the implementations of the latest engineering advances. At the same time SE solutions are easier to update but are far from user-friendly and are difficult to migrate between installations.

Deep neural network-based machine intelligence methods have brought about a revolutionary advance in the field of computer vision, by allowing for learning of complex morphologies in a highly generalized fashion (*Krizhevsky et al., 2012*; *LeCun et al., 2015*). To date, these methods have not been adapted for the field of host–pathogen interaction. Typically, HCI based fluorescent imaging data from a host–pathogen interaction experiment is analyzed by classical image segmentation (*Osaka et al., 2012*; *Schmutz et al., 2013*; *Kühbacher et al., 2015*; *Ovalle-Bracho et al., 2015*). Occasionally segmentation combined with machine learning based on calculated features has been employed (*Kreibich et al., 2015*). Most of these analysis pipelines make use of open-source programs tailored with additional coding by the user to suit their specific needs and are not published in their final form as a universal open-source solution. A major short-coming of these classical image segmentation and machine learning analysis methods is that they fail at the level of quantifying host protein recruitment to the pathogen. This is largely due to the fact that traditional algorithms quantify phenotypes in a rule-based manner, using bulk statistical properties of microscopy images or their segments. Conversely, deep neural networks make use of complex patterns (e.g. shapes) within the dataset to learn phenotypes and their diversity. The neural network derives these patterns in an automatic fashion from expert-labelled data. Thus, using pattern complexity to refine classification (*Krizhevsky et al., 2012*), deep neural networks improve the biological relevance of the phenotypic readouts.

While some proprietary solutions have been employed to extract host protein recruitment data, these are impractical and insufficient for most researchers as they are tied to single and expensive microscopes and do not operate at human capacity (*Polajnar et al., 2017*; *Touquet et al., 2018*). To date, for the analysis of host protein recruitment to pathogens, artificial intelligence-driven automated analysis is neither available as an open-source nor as a commercial package. Thus, there remains a need for an open-source, intuitive, flexible, and trainable host–pathogen interaction analysis software that performs at the level of human analytic capacity (*Russakovsky et al., 2015*; *He et al., 2015*; *Haberl et al., 2018*).

Here we present a high-throughput, high-content, single-cell image analysis pipeline that incorporates machine learning and a deep convolutional neural network (CNN) ensemble for Host Response to Microbe Analysis (HRMAn; https://hrman.org/). To assure its broad applicability to infection biology, HRMAn is based on the data integration environment KNIME Analytics Platform (*Berthold et al., 2008*). The analysis relies on training of machine learning algorithms and deep neural networks that can be tailored to individual researchers' needs.

## Results

### Architecture of the high-content image analysis pipeline for analyzing host–pathogen interaction

The HRMAn pipeline (*Figure 1*), is designed to work with all file types acquired on any HCI platform or fluorescence microscope. Plate maps including experimental layouts, sample groups and replicates can be loaded, enabling HRMAn to automatically cluster results and perform error analysis. Once fed into the HRMAn pipeline, images are automatically pre-processed and clustered based on user-defined parameters (i.e. imaging specifications) and corrected for illumination. The subsequent image analysis proceeds in two stages: in stage 1, HRMAn segments images into pathogen and host cell features for single cell analysis. It then classifies these features using a decision tree learning algorithm previously trained on an annotated dataset. In stage 2, HRMAn analyzes host cell features associated with the pathogens using a CNN HRMAlexNet (derived from AlexNet architecture) trained to distinguish complex phenotypic patterns of host-protein recruitment (*Krizhevsky et al., 2012*). Robust classification is achieved by passing segmented regions of interest through multiple non-linear convolutional filters to identify characteristic phenotypic details.

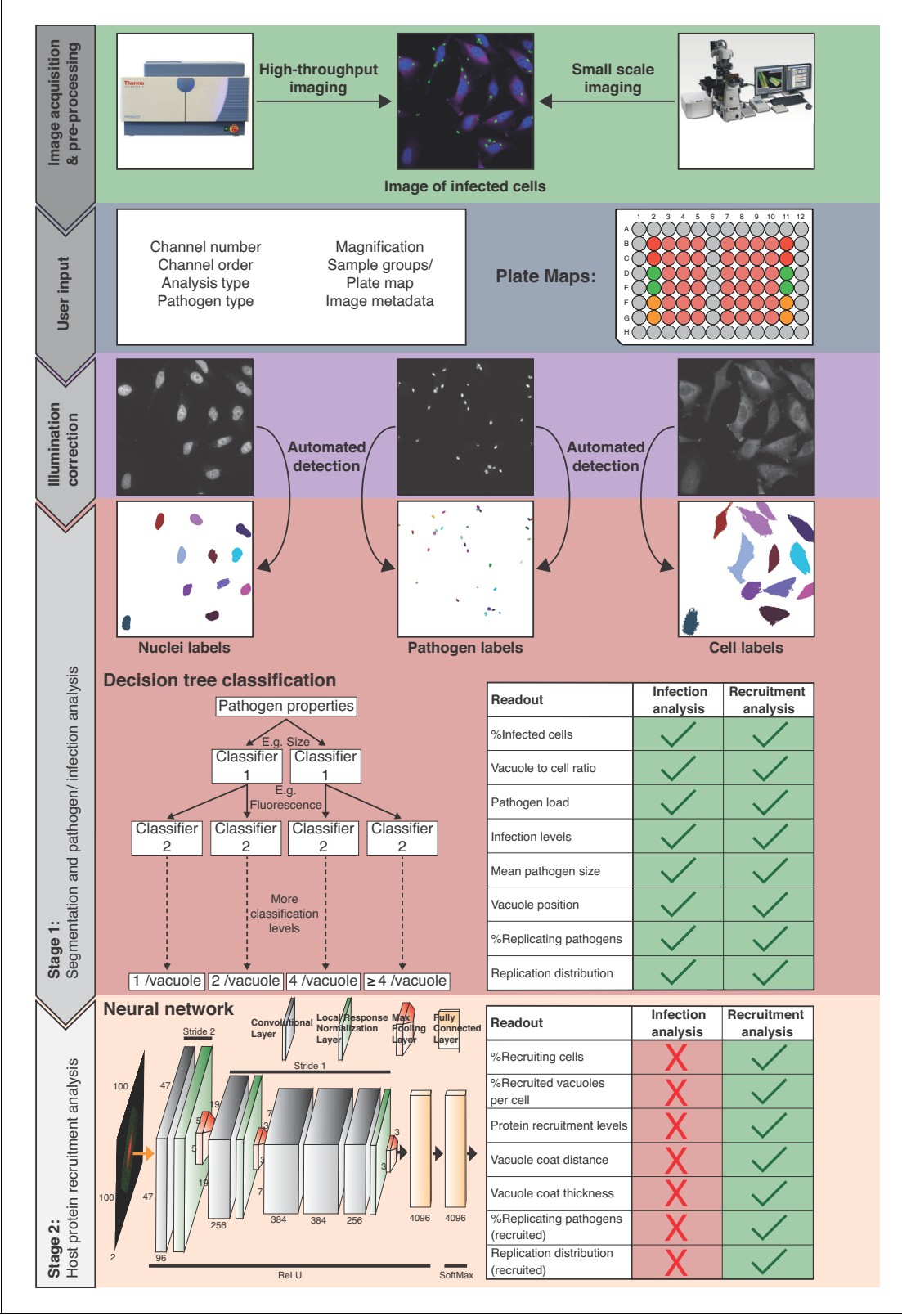

**Figure 1.** Overview of the HRMAn pipeline. Following image acquisition, on a high-content imaging platform or any other fluorescence microscope, the images can be loaded into the HRMAn software. First, the data is pre-processed and clustered based on user-defined parameters and provided plate maps. Images then undergo illumination correction and automated segmentation using Huang algorithm. Segmented images are used by a deep convolution neural network (CNN) and other machine learning based algorithms to analyze infection of cells with intracellular pathogens. Depicted is

*Figure 1 continued on next page*

*Figure 1 continued*

the CNN diagram representing two-dimensional convolutional filters with respective width, height and depth designated on filters facets. Respective change of stride in the groups of hidden layers is depicted above the diagram, while respective activation functions below the diagram. Finally, the data is written as a single file and will provide the researcher with more than 15 different readouts that describe the interaction between pathogen and host cell during infection. HRMAn is based on the open-source data integration environment KNIME Analytics Platform making it modular and adaptable to a researcher's needs. The analysis is based on training of the machine learning algorithms generating high flexibility, which can be tailored to the needs of the user.

DOI: https://doi.org/10.7554/eLife.40560.002

Finally, data are output as a single spreadsheet file providing the researcher with $\geq$15 quantitative descriptions of a pathogen and its interaction with host factors at population and single cell levels (*Figure 1*; Readouts). Importantly, by separating the analysis, HRMAn offers researchers the flexibility to perform fast, simple quantitative analysis of infection parameters using stage 1, without analyzing host protein recruitment.

## Machine learning and a convolutional neural network drive classification of pathogen replication and host defense

To train for detection and analysis of host–pathogen interactions, HRMAn was provided an annotated dataset of host cells infected with an eGFP-expressing version of the parasite *Toxoplasma gondii* (*Tg*) and stained for various host cell features (*Figure 2A*) (*Seibenhener et al., 2004*; *Clough et al., 2016*). For stage one pathogen detection and enumeration training a simplistic ML strategy – decision tree performed remarkably well. Over 35,000 *Tg*-vacuoles were analyzed by decision tree, gradient boosted tree, and random forest machine learning classification algorithms and cross-validated (*Figure 2B*). As each performed equally, a simple decision tree with Minimum Description Length (MDL) pruning, to limit overfitting, was employed for speed and accuracy of pathogen detection (>99.5% for *Tg*). Using these parameters, in addition to the readouts from stage 1 (see *Figure 1*), HRMAn detected and quantified *Tg*-containing vacuoles harbouring 1, 2, 4 or >4 fluorescent *Tg* (*Figure 2C*).

For stage 2, host protein recruitment, the CNN was trained for ubiquitin and p62 recruitment using segmented *Tg* vacuoles defined in Stage 1. Robust classification of host protein recruitment was achieved by passing these regions of interest through multiple non-linear filters to identify and differentiate between no recruitment, recruitment, and analysis artefacts (*Figure 2D*). Training over 80 epochs with negative log likelihood as a loss function, the deep CNN achieved 92.1% classification accuracy confirmed by expert-based cross-validation. Precision for 'no recruitment', 'recruitment', and 'artefacts' classes was 0.92, 0.92 and 0.71, while recall was 0.94, 0.89 and 1 respectively, hence achieving the accuracy of a human operator and far exceeding human capacity (*Figure 2E*).

To assure that uninvaded *Tg* parasites do not skew the data, stringent synchronization of infection by centrifugation and washing procedures were employed. In a pilot experiment (*Figure 2—figure supplement 1*), staining with the *Tg* vacuole marker GRA2 (*Figure 2—figure supplement 1A–B*) revealed that more than 98% of all parasites captured in the images have successfully invaded and established a PV, irrespective of the *Tg* strain used for infection (*Figure 2—figure supplement 1B*). Using a multiplicity of infection (MOI) of 3 for experiments resulted in up to 90% type I and 80% type II *Tg* infected host cells (*Figure 2—figure supplement 1C*). In line with this, we often observed that a single host cell can contain more than one PV.

## HRMAn allows for accurate high-throughput analysis of the host defense response to Toxoplasma

To demonstrate the ability of HRMAn and to expand how researchers define and classify host–pathogen interactions, the impact of IFNγ on *Tg* replication and ubiquitin/p62 recruitment to *Tg* vacuoles was analyzed (*Figure 3*).

Previous reports indicate that HeLa cells restrict the growth of *Tg* through ubiquitination of parasitophorous vacuoles and subsequent non-canonical, p62-dependent autophagy (*Selleck et al., 2015*; *Clough et al., 2016*). HeLa cells infected with eGFP *Tg* ±IFNγ were fixed 6 hr post-infection (hpi) and stained with Hoechst (nuclei) and antibodies directed against ubiquitin and p62. A total of

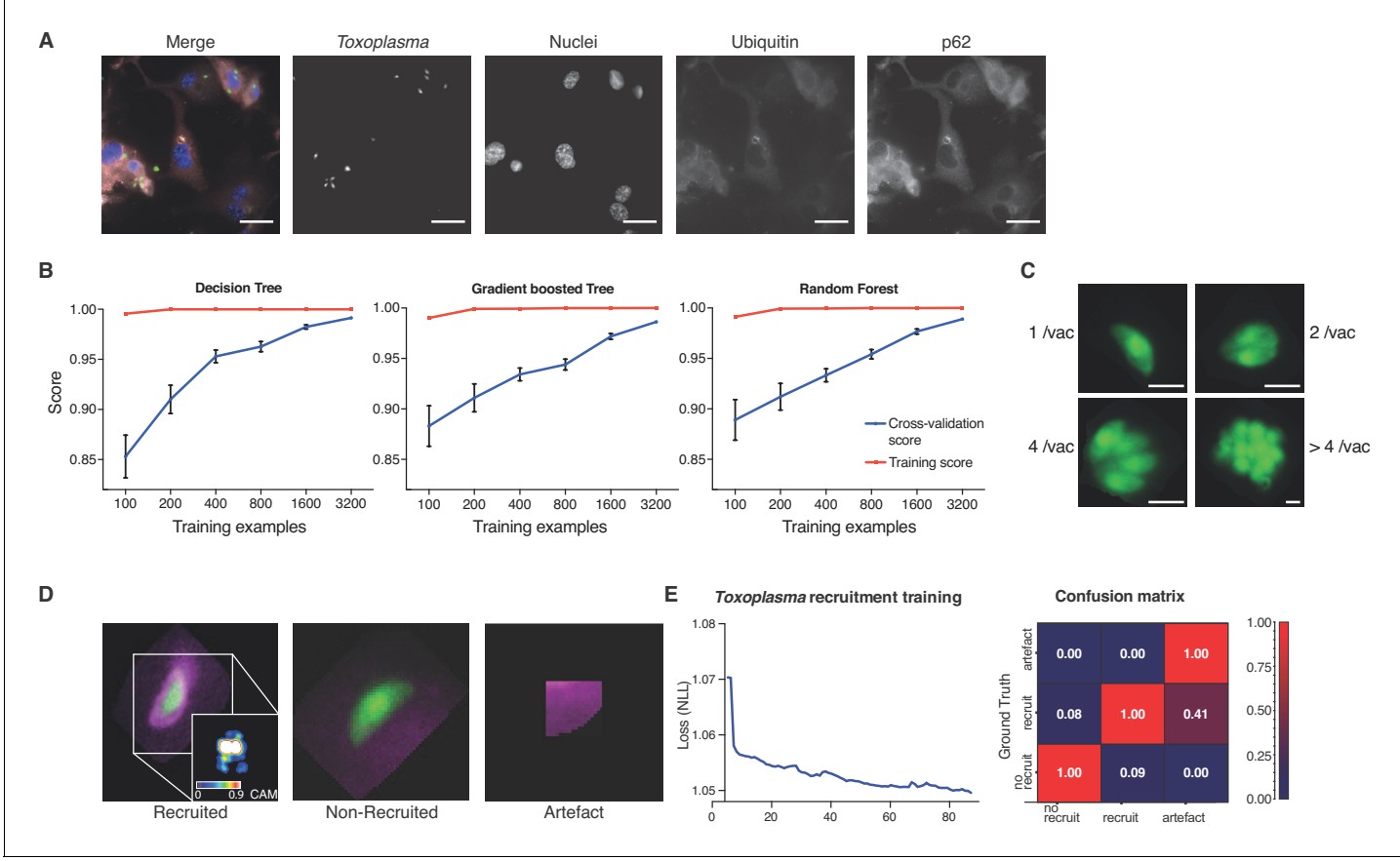

**Figure 2.** Decision-tree and convolutional neural network training for pathogen replication and host defense protein recruitment analysis. (A) Example images from one field of view. A composite image of all channels (blue: nuclei, green: *Tg*, red: Ubiquitin, grey: p62) and the single channel images are shown. Scale bar indicates a distance of 30 µm. (B) Training and cross-validation of different machine learning classification algorithms to predict parasite replication. (C) Example images of different vacuoles with the resulting classification of a trained decision tree classifier. Scale bar, 5 µm. (D) Resulting classification of the trained deep convolution neural network (CNN) with example vacuoles. For the recruited classification a class activation map (CAM) is depicted to illustrate the focus of the CNN. (E) Decrease of negative log likelihood (NLL) used as loss function during CNN training over training cycles (epochs) for *Toxoplasma gondii* model (left) and confusion matrix of *Toxoplasma gondii* model validation illustrating classification accuracy of labelled data unseen by the model, classification accuracy (0 to 1) during validation is colour-coded blue to red and indicated in the figure (right).

DOI: https://doi.org/10.7554/eLife.40560.003

The following figure supplement is available for figure 2:

**Figure supplement 1.** Infection of HeLa cells with *Toxoplasma gondii* at 6 hr post-infection.

DOI: https://doi.org/10.7554/eLife.40560.004

1,350 4-colour images were acquired on an automated microscope and loaded into HRMAn for analysis.

HRMAn automatically detected and analyzed more than 15,000 HeLa cells resulting in 15 quantitative outputs of host–pathogen interaction (*Figure 3*). Population level readouts from stage one indicated that IFNγ treatment did not impact the percentage of infected cells but decreased the number of vacuoles within host cells as well as the number of parasites per cell (*Figure 3A*). As eGFP fluorescence is lost when parasites are killed, a reduction in the ratio between vacuoles and host cells serves as an indirect measurement for parasite killing. At the single cell level, HRMAn found that IFNγ treatment resulted in a significant reduction of vacuoles per cell and a minor reduction in mean vacuole size, without impacting vacuole position (*Figure 3B*). Concomitant with this reduction in vacuole size, both the percentage of replicating parasites, and the number of parasites per vacuole were significantly reduced by IFNγ treatment (*Figure 3C*). Thus, IFNγ−mediated control

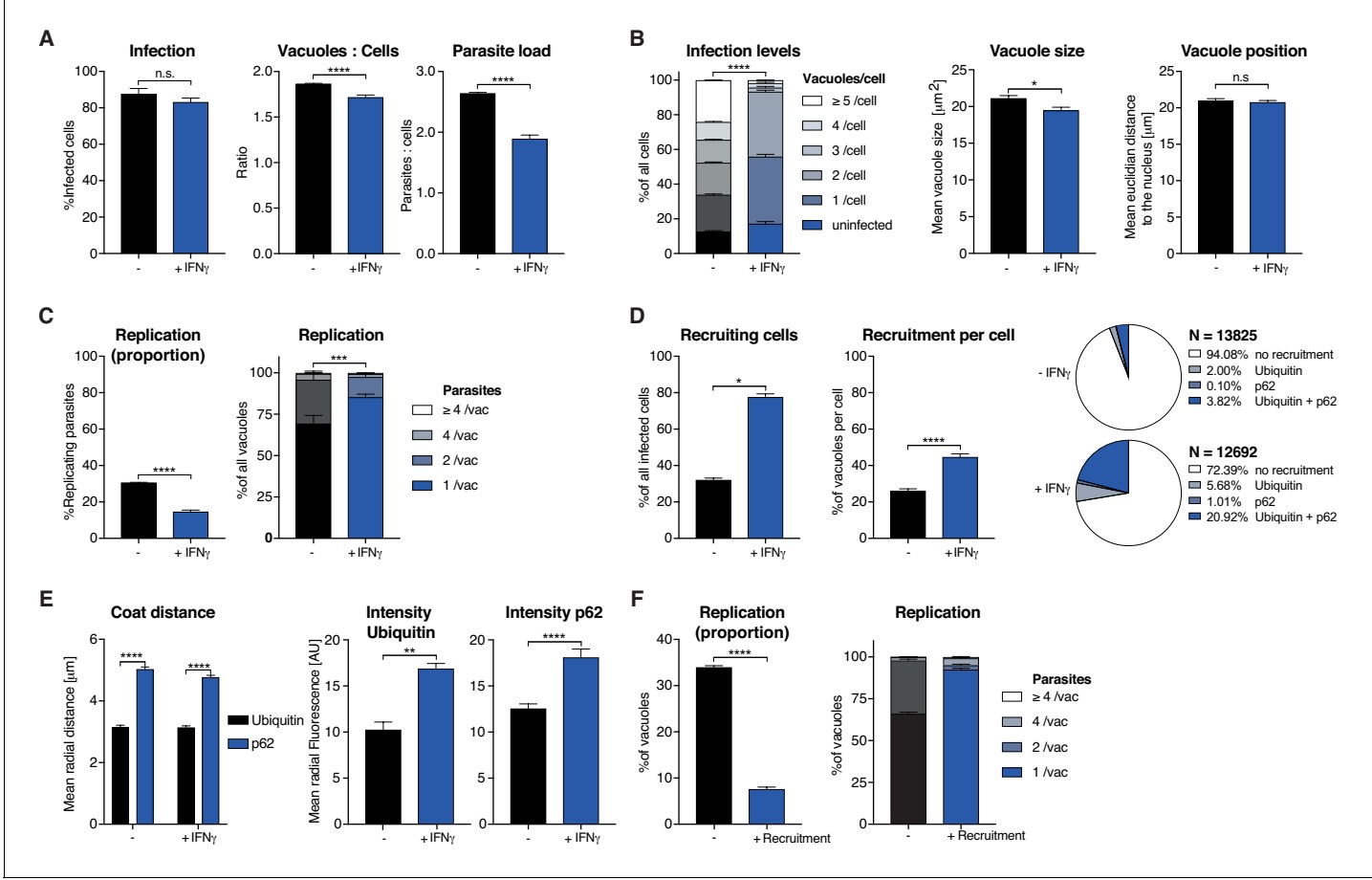

**Figure 3.** Analysis of *Toxoplasma gondii* infection in IFNγ-treated HeLa cells. HeLa cells were stimulated with 100 IU/mL IFNγ, infected with type I (RH) *Toxoplasma gondii* (*Tg*) and analyzed 6 hr post-infection. (**A**) Infection parameters depicted as total percent of *Tg* infected cells, the ratio of *Tg* vacuoles to cells and the ratio of parasites to cells. (**B**) Cellular readouts showing the proportion of cells that contain a varying numbers of parasite vacuoles, the mean vacuole size of *Tg* and the vacuole position as the value of the mean Euclidian distance of *Tg* vacuoles to the host cell nucleus. (**C**) Replication capacity of *Tg* shown as the proportion of replicating parasites and the distribution of replicating *Tg*. (**D**) Cellular response to infection with *Tg* measured as the percentage of cells that decorate vacuoles and the average proportion of vacuoles per cell that are being decorated simultaneously and the overall proportion of ubiquitin and/or p62 decorated *Tg* vacuoles. N shows the total number of vacuoles analyzed for each condition, percentages are indicated in the legend. (**E**) Properties of the host protein coat on *Tg* vacuoles as the average coat distance for ubiquitin and p62 to *Tg* and mean fluorescence intensity of ubiquitin and p62 at *Tg* vacuoles. (**F**) Fate of *Tg* vacuoles grouped based on host protein recruitment. The proportion of replicating parasites and the replication distribution based on recruitment status of the vacuole are shown. All data shown above represent the mean of N = 3 experiments±SEM. Significance was determined using unpaired t-tests, n.s. = not significant, *p≤0.05; **p≤0.01, ***p≤0.001, ****p≤0.0001.

DOI: https://doi.org/10.7554/eLife.40560.005

The following figure supplements are available for figure 3:

**Figure supplement 1.** IFNγ dose-dependent killing and replication-inhibition of *Toxoplasma gondii* in five human cell types at 24 hr post-infection.

DOI: https://doi.org/10.7554/eLife.40560.006

**Figure supplement 2.** IFNγ dose-dependent replication-inhibition of *Toxoplasma gondii* in five human cell types analyzed as parasites per vacuole at 24 hr post-infection.

DOI: https://doi.org/10.7554/eLife.40560.007

**Figure supplement 3.** Systematic analysis of IFNγ-dependent cellular control of *Toxoplasma gondii* infection of 5 human cell types at 6 hr post-infection.

DOI: https://doi.org/10.7554/eLife.40560.008

**Figure supplement 4.** Systematic analysis of IFNγ-dependent replication control of *Toxoplasma gondii* infection of 5 human cell types at 6 hr post-infection.

DOI: https://doi.org/10.7554/eLife.40560.009

**Figure supplement 5.** Systematic analysis of IFNγ-dependent replication control of *Toxoplasma gondii* infection of 5 human cell types at 6 hr post-infection analyzed as parasites per vacuole.

*Figure 3 continued on next page*

*Figure 3 continued*

DOI: https://doi.org/10.7554/eLife.40560.010

**Figure supplement 6.** Ubiquitin and p62 host protein recruitment to *Toxoplasma gondii* type I and II vacuoles in 5 IFNγ-treated human cell lines at 6 hr post-infection.

DOI: https://doi.org/10.7554/eLife.40560.011

**Figure supplement 7.** Characterization of the effect of host protein coating of *Toxoplasma gondii* type I and II vacuoles in 5 IFNγ-treated human cell lines at 6 hr post-infection.

DOI: https://doi.org/10.7554/eLife.40560.012

of *Tg* in HeLa cells involves both parasite-killing and restriction of *Tg* replication. Importantly, HRMAn offers a wide range of readouts in stage 1 analysis allowing for more detailed information on the dynamics of infection and clearance than typically seen with manual counting. To allow the user to decide which readouts are best suited to answer their specific research question some redundancy has been purposely built in (e.g. mean vacuole size vs. % Replicating). For example, here we focused on parasites per vacuole and the proportion of infected cells, as opposed to the number of individual vacuoles per host cell.

In stage 2, analysis of the >25,000 vacuoles identified in stage 1, showed that the number of host cells with ubiquitin/p62-positive vacuoles and the percentage of ubiquitin/p62-positive vacuoles per host cell increased with IFNγ (*Figure 3D*). Distribution analysis indicated that in untreated cells, only 5.92% of vacuoles were decorated with ubiquitin, p62, or both. This number rose to 27.61% in IFNγ-treated cells, the majority of which (20.92%) were double-positive for ubiquitin/p62 (*Figure 3D*). By quantifying the radial fluorescence intensity distribution of these host factors, HRMAn revealed that ubiquitin was more closely associated with *Tg* vacuoles than p62 and that recruitment of both was increased by IFNγ treatment (*Figure 3E*). This is in agreement with the notion that p62 binds a ubiquitinated vacuole substrate through its UBA domain (*Seibenhener et al., 2004*; *Clough et al., 2016*). Finally, by analyzing vacuoles that recruit ubiquitin/p62, HRMAn indicated that restriction of *Tg* replication occurs in vacuoles decorated with these host defense proteins (*Figure 3F*). Collectively, this data indicates that in HeLa cells, IFNγ drives both parasite killing as well as recruitment of ubiquitin/p62 to *Tg* vacuoles, which acts to restrict parasite replication (*Figure 3*). The results demonstrate the capacity of HRMAn to provide a quantitative, multi-parametric readout of host–pathogen interaction at population and single-cell levels.

As a high-throughput, high-content analysis program, HRMAn removes experimental size constraints imposed by manual quantification. To illustrate this, HRMAn was used to systematically analyze the impact of IFNγ treatment on type I and type II *Toxoplasma* strains in five human cell lines: HeLa (cervical carcinoma epithelial), PMA-differentiated THP-1 (macrophage-like), A549 (lung carcinoma epithelia), HFF (primary fibroblasts), and HUVEC (primary endothelial cells) (*Figure 3—figure supplements 1–7*).

First, stage 1 HRMAn was used to ascertain the impact of varying concentrations of IFNγ (50–500 IU/ml) on *Tg* infection, killing, and replication. (*Figure 3—figure supplement 1*). For each host cell line (*Figure 3—figure supplement 1A*), a dose-dependent reduction in *Tg* infection was seen (*Figure 3—figure supplement 1B*). Assessment of the vacuole:cell ratio and mean vacuole size indicated that THP-1s, HFFs, and HUVECs limit infection by IFNγ-dependent *Tg* killing, while HeLas and A549s do so by restricting replication (*Figure 3—figure supplement 1C–D*). Quantification of the number of parasites per vacuole indicated that HeLas and A549s acutely restrict type I and type II *Tg* replication at all concentrations of IFNγ (*Figure 3—figure supplement 2B–C*), while THP-1s, HFFs, and HUVECs are far more limited in this capacity (*Figure 3—figure supplement 2A,D–E*).

Next, HRMAn was employed on all 5 cell lines infected with either type I and type II *Tg* ±100 IU/ml IFNγ and immuno-stained for ubiquitin and p62. *Figure 3—figure supplement 3–7* display the 15 quantitative readouts compiled by HRMAn of 9000 fields of view (~90 GB) and >175,000 vacuoles identified in stage 1. Taking advantage of the large-scale capabilities of HRMAn, we found that all host cell types can mediate IFNγ-dependent type I and II *Tg* killing (*Figure 3—figure supplement 3B–C*), and growth restriction (*Figure 3—figure supplement 4A–B*) to similar levels. *Tg* vacuoles show strain-dependent (A549, HUVEC), and strain-independent (HFFs) IFNγ-stimulated movement towards the nucleus (*Figure 3—figure supplement 4C*). HRMAn revealed that type II *Tg* grew slower than type I *Tg* in each host cell line and that their growth decreased more upon treatment

with IFNγ (*Figure 3—figure supplement 5A–B*). Consistent with this, stage 2 HRMAn showed that all cell types could recruit ubiquitin and/or p62 equally well (*Figure 3—figure supplement 6A*), while a greater percentage of type II vacuoles per cell were decorated in response to IFNγ-priming (*Figure 3—figure supplement 6B*). The exception to this were THP-1 cells, which did not mount a strain-specific response (*Figure 3—figure supplement 6B*). Distribution analysis further indicated that THP-1s display a higher intrinsic capacity to decorate *Tg* vacuoles than other cell lines, even in the absence of IFNγ (*Figure 3—figure supplement 6C*). While no cell-type dependent differences in ubiquitin or p62 coat distance were observed (*Figure 3—figure supplement 7A*), THP-1s not only decorate vacuoles with more ubiquitin upon IFNγ stimulation, they also appear to recruit p62 in an IFNγ-independent fashion (*Figure 3—figure supplement 7B*). Decorated vacuoles in all host cell types displayed a greater ability to restrict the growth of type II versus type I *Tg* upon IFNγ treatment (*Figure 3—figure supplement 7C–D*). These results highlight the ability of HRMAn to provide high-throughput and quantitative single-cell analysis of host–pathogen interactions at a scale not achievable by automated bulk or manual quantification.

## HRMAn can be adapted for bacteria-host interaction analysis

To demonstrate its flexibility, HRMAn was trained to recognize the bacterium *Salmonella enterica* Typhimurium (STm) - a pathogen 16x smaller than *Tg* (0.5 µm vs. 8 µm) - and then set to analyze the impact of IFNγ on bacterial killing, replication, and ubiquitin recruitment. Stage 1 outputs showed that similar to *Tg* (*Figure 3*), IFNγ treatment in HeLa cells reduced the ratio of STm vacuoles/cell and the bacterial load, without impacting the percent of infected host cells (*Figure 4A*). At the single cell level, HRMAn found a significant reduction in the number of STm vacuoles/cell, consistent with a reduction in vacuole size, percent of replicating bacteria, and reduced numbers of STm/vacuole (*Figure 4B–C*). These results demonstrate that HeLa cells can control infection with STm through IFNγ−dependent bacterial killing and growth restriction.

For stage 2, we used the *Tg* recruitment model as input to retrain HRMAn for quantification of ubiquitin recruitment to STm (*Figure 4D*). This allowed us to achieve 69.9% classification accuracy, confirmed by expert-based cross-validation, in just 40 epochs using 10-fold less non-augmented image data (*Figure 4D*). It's known that HeLa cells restrict STm growth by maintaining vacuole integrity; the small percentage of bacteria that escape vacuoles are decorated with ubiquitin and subsequently cleared by autophagy (*Noad et al., 2017*; *van Wijk et al., 2017*). Interestingly, stage 2 HRMAn showed that the percent of host cells which recruit ubiquitin to STm doubles upon IFNγ treatment, while the percent of decorated vacuoles/cell increases only slightly (*Figure 4E*). As seen with *Tg* (*Figure 3E*, *Figure 3—figure supplement 7A*), IFNγ does not impact the distance of the ubiquitin coat to STm but increases its thickness (*Figure 4F*). This indicates that more ubiquitin is recruited to cytosolic STm in the presence of IFNγ and growth of decorated bacteria was restricted (*Figure 4G*). Consequently, although IFNγ treatment increases the number of host cells that recruit ubiquitin to STm and the intensity of that recruitment, at the single-cell level HeLa cells appear to have reached their capacity for detection and autophagy-mediated clearance of cytosolic/ubiquitinated STm independent of IFNγ treatment (*Figure 4E–G*).

## HRMANs versatility allows for rapid adaption to study pathogen biology

To illustrate the versatility of HRMAn and the advantage of the modular architecture combined with the accessible user interface that comes with the KNIME Analytics platform, we performed experiments to stress HRMAn's applicability and adaptability to study pathogen-driven parameters of infection. Using transgenic *Tg* lines expressing different parasite virulence factors, we were able to reproduce and expand upon published data (*Virreira Winter et al., 2011*). We confirmed that expression of ROP16 from type I *Tg* or lack of GRA15 in otherwise isogenic type II parasites (PruA7) reduces the recruitment of murine guanylate binding proteins (Gbps) to the vacuoles. Similarly, expression of type I ROP18 in type III parasites (CEP) also reduced recruitment of murine Gbps 1, 2 and 5 compared to isogenic type III parasites (*Figure 5A*). This analysis shows that HRMAn can be used to study effects of pathogen effector proteins on an established host phenotype.

Next, we asked whether HRMAn can accurately measure parasite effector proteins targeted elsewhere within infected host cells. *Tg* is known to secret multiple effector proteins upon invasion and

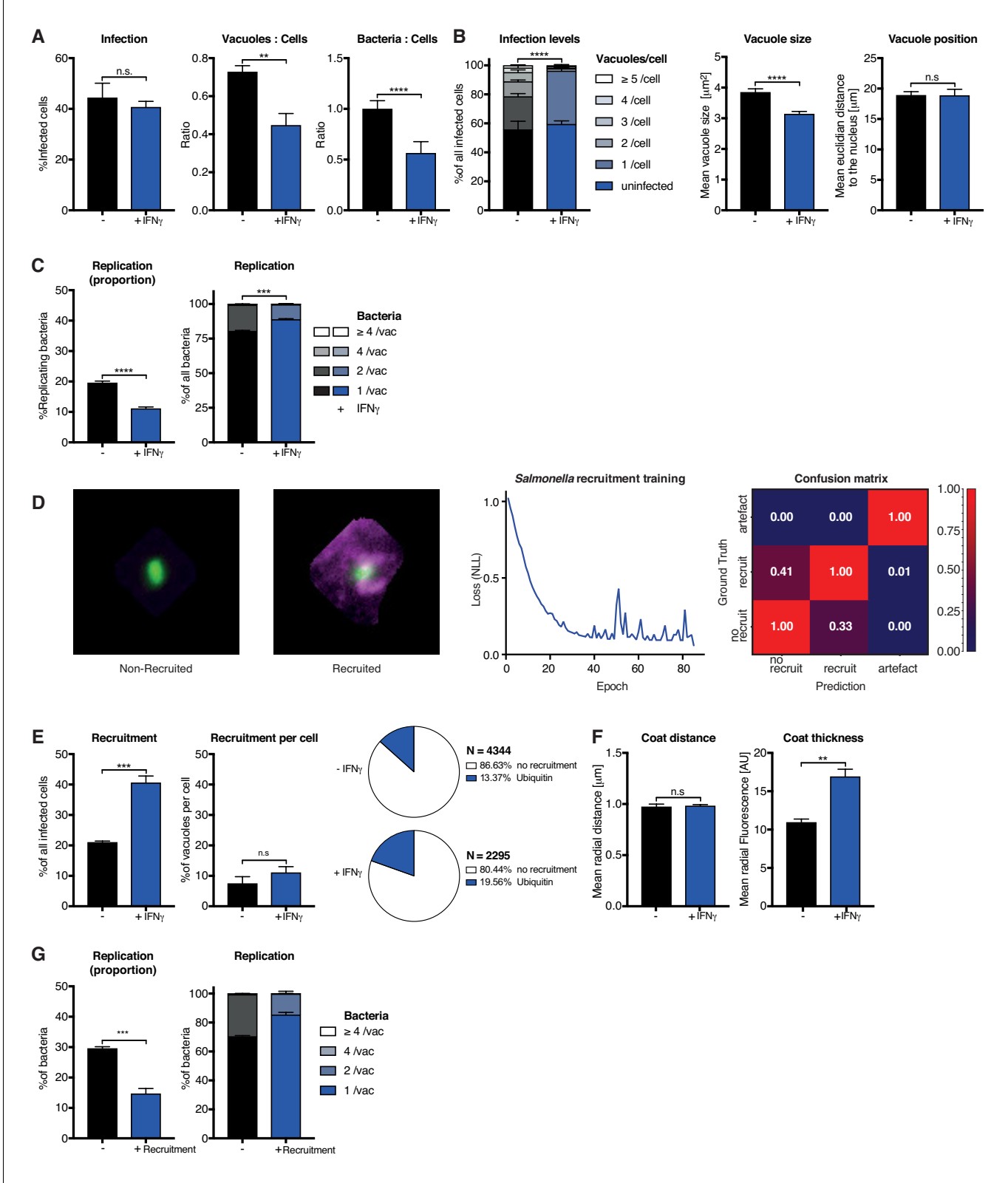

**Figure 4.** Analysis of *Salmonella enterica* Typhimurium infection in IFNγ-treated HeLa cells. HeLa cells were stimulated with 100 IU/mL IFNγ, infected with *Salmonella enterica* Typhimurium (STm) and analyzed 2 hr post-infection. (A–C) Stage one infection analysis parameters. (A) Infection parameters depicted as total percent of STm infected cells, the ratio of STm vacuoles to cells and the ratio of bacteria to cells. (B) Cellular readouts showing the proportion of cells that contain a certain number of bacteria vacuoles, the mean vacuole size of STm and the vacuole position as the value of the mean

*Figure 4 continued on next page*

Figure 4 continued

euclidian distance of STm vacuoles to the host cell nucleus. (C) Replication capacity of STm shown as the proportion of replicating bacteria and the distribution of replicating STm. (D) Training of the deep convolution neural network (CNN) to analyze host protein recruitment to STm vacuoles and bacteria. Left: Example images showing the difference of no recruitment versus ubiquitin (magenta) recruitment to STm. Middle: Decrease of negative log likelihood (NLL) used as loss function during CNN training over training cycles (epochs) for STm model. Right: Confusion matrix of STm model validation, classification accuracy (0 to 1) during validation is colour-coded blue to red and indicated in the figure. (E) Cellular response to infection with STm measured through the percentage of cells that decorate vacuoles and the average proportion of vacuoles per cell that are being decorated simultaneously and the overall proportion of ubiquitin decorated STm vacuoles. N shows the total number of vacuoles analyzed for each condition, percentages are indicated in the legend. (F) Properties of the host protein coat on STm vacuoles as the average coat distance for ubiquitin to STm and mean fluorescence intensity of ubiquitin. (G) Fate of STm grouped based on host protein recruitment. Shown is the proportion of replicating bacteria and the replication distribution based on recruitment status of the vacuole. All data shown above represent the mean of N = 3 experiments±SEM. Significance was determined using unpaired t-tests, n.s. = not significant, *p≤0.05; **p≤0.01, ***p≤0.001, ****p≤0.0001.

DOI: https://doi.org/10.7554/eLife.40560.013

during replication. Examples include the parasite proteins GRA16, GRA24 and *Tg*IST, that are secreted beyond the boundaries of its PV and subsequently translocated to the host cell nucleus (*Bougdour et al., 2013*; *Braun et al., 2013*; *Gay et al., 2016*). Using HRMAn, we were able to visualize accumulation of these three *Tg* effector proteins (tagged with an HA-tag) in the nucleus of the host cell. HRMAn, further indicated that the levels of nuclear accumulation correlated to the number of parasites contained within the infected cell (*Figure 5B*). Thus, HRMAn can be employed to analyze both host and parasite parameters during infection in an unbiased, accurate and high content manner.

## Discussion

Recent advances have made deep CNNs a powerful image analysis method (*Simonyan and Zisserman, 2014*; *He et al., 2015*; *Ioffe and Szegedy, 2015*; *LeCun et al., 2015*; *Russakovsky et al., 2015*; *Haberl et al., 2018*). Inspired by abstraction of animal visual cortex architecture, CNNs are able to generalize patterns independent of minor phenotypic differences (*Hubel and Wiesel, 1968*; *Matsugu et al., 2003*). Combining automated image segmentation, machine learning and a deep CNN in an ensemble, HRMAn is a powerful open-source, user-friendly software for the analysis of host–pathogen interaction at the single-cell level. We based HRMAn on the KNIME Analytics Platform (*Berthold et al., 2008*). Being highly modular, GUI-based and user-friendly, HRMAn can rapidly be updated with latest technological advances, yet remains transparent for the average user. Furthermore, the ready-to-use DL4J library modules we employed allow for incorporation of the latest advances in the field of artificial intelligence in a click-through manner with zero coding. To date, HRMAn represents the only open-source CNN-driven host-pathogen analysis solution for fluorescent images. While HRMAlexNet is a rather simple architecture, more complex architectures can be easily implemented through recently introduced KNIME-Keras integration (*Chollet, 2015*), which may facilitate improvement in the classification accuracy. This is important, as it moves the phenotype of host defense protein recognition of pathogens into the realm of HCI at the level of artificial intelligence and thus human accuracy and capacity. Many automated image analysis programs, some of which incorporate machine learning elements, have been developed and are successfully used for classical image segmentation (*Supplementary File 1*). However, when presented with the problem of classifying host protein recruitment to a pathogen, inaccurate classical image segmentation could lead to erroneous results. Employing an artificial intelligence algorithm, HRMAn circumvents these problems and delivers user-defined automated and unbiased enumeration of this subset of the host-pathogen interplay.

Using *Tg* and STm infection models, we demonstrate that HRMAn is capable of detecting and quantifying multiple pathogen and host parameters. Importantly, we show that HRMAn can be adapted easily to two entirely different pathogens, that not only differ in size by a magnitude, but also display distinct growth rates and infection dynamics. The easy adaption of HRMAn for different pathogens and research questions will prove useful for any lab working in image-based infection biology. Designed for biologists, HRMAn requires no coding or specialized computer science knowledge. Its modular architecture and the use of KNIME, which provides a graphical representation of

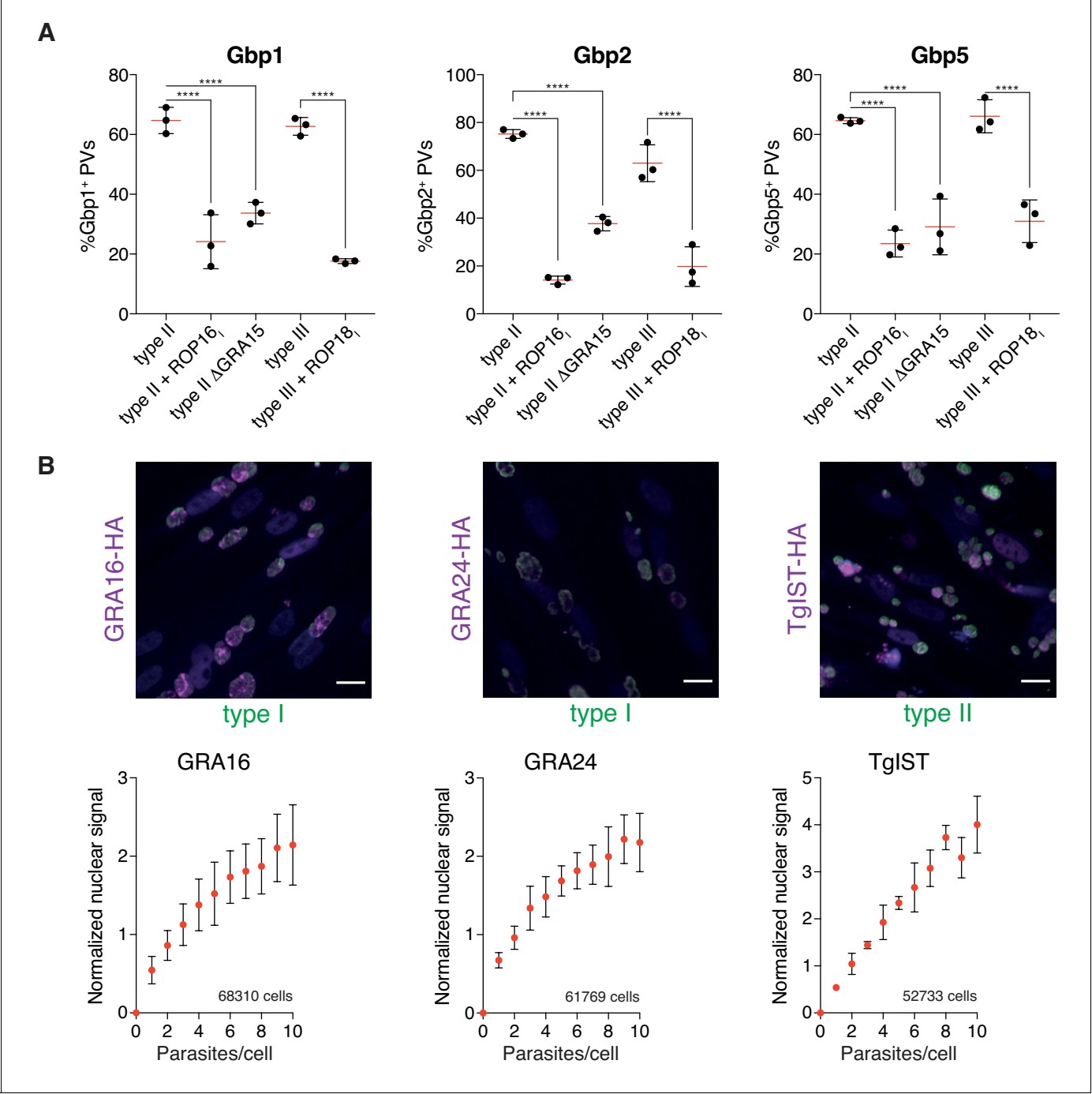

**Figure 5.** HRMAn can be adapted to study pathogen biology. (A) HRMAn-based quantification of Gbp recruitment to *Tg* vacuoles. Red lines show mean ± SEM of N = 3 experiments. (B) Quantification of *Tg* protein secretion and translocation to the host cell nucleus. HFF cells were infected with type I *Tg* expressing GRA16-HA or GRA24-HA or with type II *Tg* expressing *Tg*IST-HA and fixed after 18 hr. Secreted proteins were visualized by staining with anti-HA (magenta) and *Tg* was stained with anti-SAG1 (green). Scale bar, 20 µm. Fluorescence signal in the host cell nucleus was quantified, correlated to the number of parasites per cell and normalized to the signal of uninfected cells. Overall number of analyzed cells are indicated in the graphs. Data represented as mean ± SEM of N = 3 experiments. Significance from one-way ANOVA comparing to the respective WT, n.s. = not significant, *p≤0.05; **p≤0.01, ***p≤0.001, ****p≤0.0001.
DOI: https://doi.org/10.7554/eLife.40560.014

the analysis pipeline, allows users to tailor experimental outputs to their own datasets and questions. As the models we have generated can be used as primers to lower the training dataset size, computation power and training time requirements, HRMAn can be rapidly applied to similar large-scale, image-based experimental setups. As such, HRMAn will allow a broad range of researchers to extend into the realm of high-throughput, unbiased, quantitative single-cell analysis of host–pathogen interaction.

## Materials and methods

### Code and image availability

All open-source KNIME workflows used in this publication can be found at: https://github.com/HRMAn-Org/HRMAn and on the homepage https://hrman.org/ under GPLv3 open-source software license to allow for rapid and open dissemination and free availability for the research community. The models and their respective weights obtained through training will be deposited on GitHub and the homepage as well.

All images used to train the neural networks and other machine learning algorithms in this study are available upon request.

### Image acquisition

For simple infection analysis (stage 1), 96-well plates (see Microscopy sample generation) were imaged on an ArrayScan VtI Live High Content Imaging Platform (Thermo Scientific) using 20x magnification and depending on the experiment, 15–20 fields of view per well.

For recruitment analysis to *Toxoplasma gondii* (*Tg*) vacuoles, glass-bottom 96-well plates were imaged on an ArrayScan VtI Live High Content Imaging Platform but using 40x magnification and depending on the experiment, 50 fields of view per well. In both cases, following image acquisition, the images were exported from HCS Studio Cell Analysis Software as single channel 16-bit tiff files before they were fed into the HRMAn analysis pipeline.

For recruitment analysis to *Salmonella enterica* Typhimurium (STm) vacuoles, images of coverslips were acquired on a Ti-E Nikon microscope equipped with an LED-illumination and an Orca-Flash4 camera using a 60x magnification. 75 fields of view per coverslip were acquired using multi-position acquisition. Images were exported as single channel 16-bit tiff files with Nikon NIS Elements software before they were fed into the HRMAn analysis pipeline.

Generally, HRMAn can work with any common image file format, but the use of uncompressed, lossless formats like tiff (or png) is recommended. Furthermore, HRMAn can work with images acquired on any type of fluorescence microscope and is truly platform independent.

### Image analysis using HRMAn

Following image acquisition, the images were loaded into the HRMAn pipeline. Images can be in any common file format, preferably as single-channel tiff files. The used image reader loads images from all file formats supported by SCIFIO (more information can be found here: http://scif.io). If the images were not acquired on a high-content imaging platform, they can be renamed with HRMAn to mimic the file names and the plate format. This is needed to cluster the output data and perform error calculation. Furthermore, the OME-XML-metadata is loaded and information on the image is extracted (e.g. image size, type and origin). While the images are loaded into KNIME, the user is asked to provide some basic information on the image acquisition and the type of analysis to be performed. This includes the used magnification, type of analysis, channel number and order and providing a plate map to cluster the data.

HRMAn then pre-processes the images, metadata and provided information and lets the user inspect the input images arranged into a grid and sorted by the field of view. Next the input images undergo illumination correction by dividing the background as a mean image of all acquired images in a channel-wise fashion. Following this step, the individual channels are segmented to detect the Nuclei, the pathogens and the cells:

1. Nuclei are detected using Otsu's method thresholding (*Otsu, 1979*), a watershed and connected component analysis. Fields of view containing insufficient numbers of nuclei (i.e. empty fields) are excluded from the following analysis.

2. The pathogens (or vacuoles) are detected after image normalization and filtering through thresholding using Otsu's method. Incomplete labels are corrected by filling holes and pathogen vacuoles are separated through water-shedding. Labels are created with a connected component analysis.

3. Cell labels are created using Huang thresholding (*Huang et al., 2004*) and a Voronoi segmentation using the nuclei labels as starting points. Optionally the images can be enhanced using Contrast Limited Adaptive histogram equalization (CLAHE) to improve the segmentation accuracy. All cell labels touching the border of an image are excluded from the analysis.

Furthermore, the created labels are filtered based on their size and the user-defined parameters such as magnification and detector size. The filter values for STm and *Tg* were empirically determined using thousands of images from different experiments. Based on which pathogen type is chosen by the user, HRMAn will adjust the filters automatically. Using labelling arithmetic, pathogen labels that are not contained within a cell label are removed from the dataset, as they represent extracellular pathogens. The created labels can then be inspected by the user through an interactive label viewer displaying the original image next to the labels.

Using the created labels, the infection readouts for stage 1 are created: First, cell numbers $N_{Cells}$ and vacuole numbers $N_{Vacuoles}$ are determined by counting the numbers of respective labels in all acquired fields per well (=replicate). Using these values, the vacuole to cell ratio is calculated:

$$Vacuole{:}Cells = \frac{N_{Vacuoles}}{N_{Cells}}$$

Next, the dependencies between the cell labels and vacuole labels are used to calculate the proportion of infected cells and the infection levels of the cells. The label dependencies determine which vacuole labels $V_1$, $V_2$, ..., $V_i$ are contained by which cell label $C_1$, $C_2$, ..., $C_j$. If a cell label C contains at least one vacuole label V, the cell is considered as infected cell $C_{inf}$. This is used to calculate the proportion of infected cells:

$\%Infected\ cells = \frac{N_{Cinf}}{N_{Cells}}$, with $N_{Cinf}$ as the number of infected cells $C_{inf}$

To determine the more precise distribution describing how many vacuoles are contained by which proportion of cells (=Infection levels) the cells C are split into subgroups according to the number of vacuoles they contain (no vacuoles or uninfected = $C_0$, 1 vacuole = $C_1$, ..., 5 or more vacuoles per cell = $C_{\geq 5}$) and then the proportion is calculated:

0 vacuoles per cell (uninfected): $uninfected = \frac{N_{C0}}{N_{Cells}}$

1 vacuole per cell: $1\ vac/\ cell = \frac{N_{C1}}{N_{Cells}}$

...

5 or more vacuoles per cell: $>5\ vac/\ cell = \frac{N_{C\geq5}}{N_{Cells}}$

Also based on the dependencies, the mean euclidean distance $d$ between the centroid of the vacuole labels (with coordinates $X_V$ and $Y_V$) within a cell and its nucleus' centroid (with coordinates $X_N$ and $Y_N$) is determined as the position of the vacuole:

$$d = \sqrt{(X_V - X_N)^2 + (Y_V - Y_N)^2}$$

Using the vacuole labels, of vacuoles contained within cells, and working on the original images the properties of the vacuoles are measured as mean values for each well. These include mean vacuole size, shape descriptors (Circularity, Perimeter, Convexity, Extent, Diameter) and fluorescence properties (Minimum, Mean and Maximum Fluorescence).

Using the above determined values as attributes, a decision-tree machine learning algorithm determines good classifiers and employs them to classify each vacuole label $V_i$ based on how many individual pathogens $P_{Vi}$ it contains. This step requires providing an annotated dataset.

Based on this classification the vacuoles can be divided into individual groups for the number of pathogens they contain (1/ vac = $V_{1Vac}$, 2/ vac = $V_{2Vac}$, 4/ vac = $V_{4Vac}$ and four or more/vac = $V_{\geq 4Vac}$) and the number of vacuoles in each group is counted (e.g. number of vacuoles that contain just one pathogen = $N_{V1Vac}$). To calculate the proportion of replicating pathogens, the number of vacuoles that contain at least two pathogens is divided by the total number of vacuoles:

$$\%Replicating = \frac{N_{V2Vac} + N_{V4Vac} + N_{V \geq 4Vac}}{N_{Vacuoles}}$$

Similarly, the individual proportions of the vacuole groups are calculated to illustrate pathogen replication distribution:

1 pathogen per vacuole: $1/\ vac = \frac{N_{V1Vac}}{N_{Vacuoles}}$

2 pathogens per vacuole: $2/\ vac = \frac{N_{V2Vac}}{N_{Vacuoles}}$

. . .

4 or more pathogens per vacuole: $>4/\ vac\ = \frac{N_{V \geq 4Vac}}{N_{Vacuoles}}$

Combining the information on the number of vacuoles and the number of pathogens $P_{Vi}$ each individual vacuole $V_i$ contains, the total number of pathogens $N_{Pathogens}$ is calculated:

$$N_{Pathogens} = \sum V_i \times P_{Vi}$$

This can be used to determine the Pathogen load by normalization to the cell number:

$$Pathogen\ Load = \frac{N_{Pathogens}}{N_{Cells}}$$

This concludes stage 1 infection analysis performed by HRMAn. In the end of the analysis, the values calculated for each well or replicate is combined with the values for the other wells belonging to the same sample group based on the user-provided plate map and error calculation is performed. If the user decides to perform only stage 1 infection analysis the HRMAn image analysis pipeline will stop here, if host protein recruitment analysis is to be performed the data will be fed into the second stage for which the implemented deep Convolutional Neural Network (CNN) has to be created and trained first.

## Deep learning setup and neural network architecture

To classify pathogen recruitment, we employed a deep Convolutional Neural Network (CNN) HRMAlexNet (*Figure 1*) inspired by published AlexNet (*Krizhevsky et al., 2012*). Our architecture consisted of a total of 5 convolutional layers, where the first two were immediately followed by local response normalization layers and max pooling layers. The last three convolutional layers were followed by one max pooling layer connected to a fully connected layer. All these layers used rectified linear unit (ReLU) as activation function (*Nair and Hinton, 2010*). To ensure our neural network can be implemented by other researchers with no coding, it was based on the open source DeepLearning4J library implementation in KNIME Analytics Platform. Changes to the original AlexNet architecture were partially introduced by the KNIME AG team and partially by us in the process of optimizing the architecture to the fluorescence microscopy data. Here further improvements in architectures may be made through recently introduced KNIME-Keras integration (*Chollet, 2015*).

While having the same number of convolutional layers as AlexNet, as introduced by the KNIME team, their dimensions are different. HRMAlexNet has two fully connected layers instead of three, which pushes the fitting burden more to the convolutional layers (*Figure 1*).

We have changed the input layer dimensions to fit the multichannel fluorescence microscopy data, since our input two channels, rather than the standard RGB dimension of three. Hence, unlike the original AlexNet, HRMAlexNet was suited to take 100 by 100 by two pixels images as input and designed to run on a single graphic processing unit (GPU).

Furthermore, we have modified the SoftMax (*Bishop, 2006*) output layer of the architecture to fit our dataset and classification problem. While optimizing the network's hyperparameters, we found that learning rate updating algorithm proposed by default in the KNIME architecture, as well as in Krizhevsky et al. - Stochastic Gradient Descent (SGD) with Nesterov momentum - was failing to converge on our fluorescence microscopy data (*Krizhevsky et al., 2012*). Therefore, we have substituted this algorithm for the more advanced learning rate update algorithm ADAM (*Kingma and Ba, 2014*) with both mean and Variance decay parameters, which proved to converge well on various data and initial learning rate ranges.

The choice of DeepLearning4J as a deep learning library allowed us to use 16-bit microscopy images directly, preventing information loss upon conversion of scientific imaging data. Our deep learning hardware was based on a single Nvidia 1080 Ti GPU set up in Intel Core i7 4790K system equipped with 32 Gb of RAM and a SSD.

## Neural network training and hyperparameters optimization

Our neural networks were trained using SGD based backpropagation (using ADAM updater with ADAM Mean Decay of 0.9 and ADAM Variance Decay of 0.999 (*Kingma and Ba, 2014*)) on the augmented original data over at least 80 epochs. To fully utilize the GPU capacity, training was performed in mini-batches of 200. We employed the Xavier algorithm for the weight initialization strategy and negative log likelihood (NLL) as our loss function (*Kingma and Ba, 2014*). We used learning rates between 0.001 and 0.01 adjusted accordingly to ensure the optimal loss curve decay during training. Together with the weights initialization strategy and the updater choice these were the main hyperparameters optimized in multiple iterations to ensure good training process.

To visualize the attention of the trained HRMAlexNet, we have created a class activation map (CAM) based on a test image from 'recruitment' class (*Figure 2*). Since HRMAlexNet architecture does not have a Global Average Pooling layer to visualize the CAM we have used the occlusion technique (*Zeiler and Fergus, 2013*).

## Data preparation, Augmentation and Model Validation

Vacuole images used for creation of our deep learning model were segmented from large field of view micrographs obtained from high-content imaging. To ensure the dimension of the images are uniform, we padded all vacuole with zero-value padding to a uniform 100 by 100 pixels size. Next, we manually labelled the segmented vacuoles into recruited, non-recruited and artefactual (in case of erroneous segmentation of the vacuole). This labelled dataset was then split into the training and test datasets.

To ensure our neural network has sufficiently diverse learning data, upon splitting the original labelled dataset into training and test subsets we performed data augmentation using a custom developed macro for ImageJ. During the augmentation processing, the original labelled dataset was concatenated with a modified version of it. The modifications included various rotations, image reflecting, image translation within the field of view. As microscopy data are typically rotation-, translation- or reflection-invariant, such modification allowed us to create a better dataset aiming at a more generalized model.

Model validation was performed using the non-augmented test fraction of the labelled dataset previously unseen by the model. For this, we used the trained model as first input and passed the labelled test data through the classifier in the second input. The classification accuracy was assessed by accuracy score, numbers of true positive, false positive, true negative and false negative, as well as Cohen's kappa values. A direct summary of the accuracy was visualized in a confusion matrix illustrating a mismatch between original label (Ground Truth) and the class assigned by the classifier (*Figure 2*).

## Host protein recruitment analysis in HRMAn

For recruitment analysis, the vacuole labels created in stage 1 of the analysis are dilated over 20 iterations to create non-overlapping regions of interest (ROIs) around them. Simultaneously, the fluorescence images of the pathogen and the respective channel with fluorescence signal of the host protein are merged into a dual channel image. The created ROIs are used to crop the dual channel images, which creates images of the pathogen and its surrounding stained host protein. The images are clipped to 100 by 100 pixel and fed into a feedforward predictor (classification) which uses the provided and trained deep convolution neural network (CNN) to classify the pathogens vacuoles based on their coating.

Once the vacuoles are separated into two groups, they are analyzed with the above described methods of stage one infection analysis but additionally comparing recruited versus non-recruited vacuoles. Thus, in addition to the overall infection parameters from stage one the user is provided with the same parameters but further layered for the cellular response.

In the case of co-recruitment analysis, two images of each vacuole are created with both containing the pathogen signal, but each containing a different second channel, representing the different stainings. After classification with the CNN, the vacuoles can be compared for recruitment or co-recruitment and all the above described parameters are calculated for them individually. Using the previously determined label dependencies of vacuoles $V_i$ and cells $C_i$ and the classification of the vacuoles $V_i$ by the CNN, HRMAn can furthermore calculate the proportion of cells that do respond to infection by decorating at least one vacuole and the proportion of vacuoles decorated per cell, if a single cell contains more than one pathogen vacuole.

Furthermore, working only on the decorated vacuoles, we used a custom-made Fiji code to create a pixel-wise radial intensity profile starting from the pathogen centroid. The distance of the maximum fluorescence intensity is then used to define the distance of the coat from the pathogen centre. Moreover, the mean fluorescence intensity of the coat is determined and can be used as a readout for the amount of protein recruited to each pathogen vacuole.

Finally, all mean values and errors for the replicate conditions, as defined by the user's plate map, are calculated and written into a single spreadsheet file. Before this, the user can also define a scaling factor between pixel and actual metric values which will adjust the output values from pixel (px) to μm or to $μm^2$ respectively.

## Cell culture and cell lines

THP-1 (TIB202, ATCC; RRID:CVCL_0006) were maintained in RPMI with GlutaMAX (Gibco) supplemented with 10% FBS (Sigma), at 37 °C in 5% $CO_2$. THP-1s were differentiated with 50 ng/mL phorbol 12-myristate 13-acetate (PMA, P1585, Sigma) for 3 days and then rested for 2 days by replacing the differentiation medium with complete medium without PMA. Cells were not used beyond passage 20. Human Umbilical Vein Endothelial cells, HUVECs, (C12203, Promocell), were maintained in M199 medium (Gibco) supplemented with 30 μg/mL endothelial cell growth supplement (ECGS, 02–102, Upstate), 10 units/mL heparin (H-3149, Sigma) and 20% FBS (Sigma). Cells were grown on plates, pre-coated with 1% (w/v) porcine gelatin (G1890, Sigma) and cultured at 37 °C in 5% $CO_2$. HUVEC were not used beyond passage 6. HeLa (ECACC, Sigma; RRID:CVCL_0030), A549 (CCL-185, ATCC; RRID:CVCL_0023), mouse embryonic fibroblasts (MEF) and human foreskin fibroblasts, HFFs (SCRC-1041, ATCC; RRID:CVCL_3285), were cultured in DMEM with GlutaMAX (Gibco) supplemented with 10% FBS (Sigma), at 37 °C in 5% $CO_2$. HeLa and A549 cells were not used beyond passage 25 and HFFs were not used beyond passage 15. All cell cultures were performed without addition of antibiotics and the cells were regularly tested for mycoplasma contamination by immunofluorescence, PCR and agar test and found to be mycoplasma-negative.

## Interferon stimulation of cells

All five cell lines used in this publication were stimulated for 16 hr in complete medium at 37 °C with addition of 100 IU/mL human IFNγ (285-IF, R&D Systems) prior to infection, if not indicated otherwise.

## Parasite culture

*Tg* expressing luciferase/eGFP (type I RH, type II Prugniaud (Pru) and type III CEP), type II PruA7 (*Kim et al., 2007*), type II PruA7 + ROP16$_I$, type II PruA7 ΔGRA15 and type II CEP + ROP18$_I$ (all from *Virreira Winter et al., 2011*) were maintained in vitro by serial passage on monolayers of HFF cells, cultured in DMEM with GlutaMAX (Gibco) supplemented with 10% FBS (Sigma), at 37 °C in 5% $CO_2$. Type II *Tg* expressing *Tg*IST-HA-Flag (Pru ΔKU80 + *Tg*IST HF), type I *Tg* expressing GRA16-HA (RH ΔKU80 + GRA16 HA) and type I *Tg* expressing GRA24-HA-Flag (Pru ΔKU80 + GRA24 HF) were a gift from Mohamed-Ali Hakimi (*Bougdour et al., 2013*; *Braun et al., 2013*; *Gay et al., 2016*).

## *Toxoplasma gondii* infection

Parasites were always passaged the day before infection onto new HFFs to obtain parasites with a high viability for infection. *Tg* were prepared from freshly 25G syringe-lysed HFF cultures in 10% FBS by centrifugation at 50 x g for 3 min and transferring the cleared supernatant into a new tube and subsequent centrifugation at 500 x g for 7 min and re-suspension of the pelleted parasites into fresh complete medium. Then, the parasites were added to the experimental cells at a MOI of 3 for both

type I and type II strains. The cell cultures with added *Tg* were then centrifuged at 500 x g for 5 min to synchronize the infection. Two hours post-infection, the cultures were thoroughly washed two times with warm PBS (806552, Sigma) to remove any uninvaded parasites and fresh complete medium was added prior to culturing at 37 °C, 5% $CO_2$ for the required time.

## Bacterial culture and infection

*Salmonella enterica* Typhimurium 12023 strain containing the plasmid pFVP25.1, carrying gfpmut3A under the control of the rpsM constitutive promoter (*Valdivia and Falkow, 1996*) were grown in Luria Bertani (LB) medium supplemented with 50 µg/ml ampicillin (11593027, Gibco). Prior to infection, bacteria were grown to induce SPI-1 T3SS expression: cultures of STm were grown at 37°C in LB, diluted 1:50 into fresh LB containing 300 mM NaCl (746398, Sigma) the next morning and incubated shaking at 200 rpm until OD600 = 0.9–1.0 was reached. Bacteria were washed in medium without FBS before use. Cells were infected at a MOI of 50 and infections were synchronized by centrifuging bacteria onto the cells at 750 x g for 10 min. 15 min post infection, the cells were thoroughly washed three times with warm PBS to remove extracellular bacteria and medium containing 100 ug/mL Gentamicin (15750060, Gibco) was added for 30 min. Then, Gentamicin concentration was reduced to 10 µg/mL and cells were incubated further at 37°C, 5% $CO_2$ for the appropriate amount of time.

## Antibodies

Antibodies used in this study were rabbit pAb anti-p62 (#PM045, MBL; RRID:AB_1279301), mouse mAb anti-GRA2 (A1298, Biovision) mouse mAb anti-ubiquitin FK2 (PW8810, Enzo Lifesciences; RRID: AB_10541840), mouse mAb anti-SAG1 (home-made) and rat mAb anti-HA (11867423001, Sigma). Secondary antibodies used were Alexa Fluor 647-conjugated goat anti-rabbit (A-21245, Invitrogen; RRID:AB_141775), anti-rat (A-21247, Invitrogen; RRID:AB_141778) or anti-mouse (A-21236, Invitrogen; RRID:AB_141725), Alexa Fluor 488-conjugated goat anti-mouse (A-11001, Invitrogen; RRID:AB_2534069) and Alexa Fluor 568-conjugated goat anti-mouse (A-11004, Invitrogen; RRID:AB_141371).

## Microscopy sample generation

### Simple infection analysis

For simple infection analysis, 30,000 THP-1s per well were seeded 5 days prior to IFNγ treatment and differentiated with 50 ng/mL PMA for three days and then rested for 2 days in complete medium. HFFs were harvested by washing a confluent monolayer with PBS and subsequent lifting of the cells with 0.05% Trypsin-EDTA (Gibco). Cells were centrifuged at 250 x g for five mins, re-suspended in fresh DMEM plus 10% FBS and 20,000 HFFs per well were seeded the day before IFNγ treatment. Similarly, HUVECs were harvested and 15,000 cells per well were seeded in complete medium the day before IFNγ treatment. A549s and HeLa cells were harvested in the same way and 8,000 cells per well were seeded the morning before IFNγ treatment. All cells were seeded on 1% (w/v) porcine gelatin (G1890, Sigma) pre-coated black-wall, clear bottom 96-well plates (Thermo Scientific). In the evening, all cells were treated with 100 IU/mL IFNγ or medium and left at 37°C, 5% $CO_2$ overnight. The next morning the cells were infected with either *Tg* or STm as described above. After the appropriate infection duration, the infected cells were again thoroughly washed with warm PBS to remove as many uninvaded pathogens as possible and subsequently fixed with 4% methanol-free paraformaldehyde (28906, Thermo Scientific). Fixed specimens were permeabilized with Perm-Quench buffer (0.2% (w/v) BSA and 0.02% (w/v) saponin in PBS) for 30 min at room temperature. Then PermQuench buffer containing 1 µg/mL Hoechst 33342 (H3570, Invitrogen) and 2 µg/mL Cell-Mask Deep Red plasma membrane stain (C10046, Invitrogen) were added and samples were incubated at room temperature for 1 hr. After staining, the specimens were washed with PBS five times and kept in 200 µL PBS per well for imaging.

### Recruitment analysis

For recruitment analysis, the cells were prepared as described above, but they were seeded on 1% (w/v) porcine gelatin pre-coated black-wall, glass bottom 96-well imaging plates CG 1.0 (130-098-264, MACS Miltenyi) to allow higher resolution imaging. After fixation, cells were permeabilized identically and then stained with primary antibody diluted in PermQuench buffer for 1 hr at room

temperature. After three washes with PBS, cells were incubated with the appropriated secondary antibody and 1 μg/mL Hoechst 33342 diluted in PermQuench buffer for another hour at room temperature. Then, the specimens were washed with PBS five times and kept in 200 μL PBS per well for imaging.

In the case of recruitment analysis to STm vacuoles, the cells were seeded on 1% (w/v) porcine gelatin pre-coated 9 mm coverslips in 24-well plates. After fixation and identical staining procedure, the coverslips were mounted using 5 μL ProLong Gold Antifade Mountant (P36930, Invitrogen).

## Data handling and statistical measurements

Data was plotted using Prism 7.0e (GraphPad Inc.) and presented with error bars as standard error of the mean (SEM). Significance of results was determined by non-parametric one-way ANOVA or unpaired t-test as indicated in the figure legends. Correction for multiple comparisons was performed by controlling the False Discovery Rate with Two-Stage step-up method after Benjamini, Krieger and Yekutieli.

## Acknowledgements

We thank all members of the Frickel lab for productive discussion. We thank Mohamed-Ali Hakimi for providing transgenic *Toxoplasma* lines. We thank Valentin Bazarevsky and Stefan Helfrich for discussion and critical reading of the manuscript. This work was supported by the Francis Crick Institute, which receives its core funding from Cancer Research UK (FC001076), the UK Medical Research Council (FC001076), and the Wellcome Trust (FC001076). EMF was supported by a Wellcome Trust Career Development Fellowship (091664/B/10/Z). DF was supported by a Boehringer Ingelheim Fonds PhD fellowship. AY, JM were supported by core funding to the MRC Laboratory for Molecular Cell Biology at University College London (JM), the European Research Council (649101-UbiProPox), the UK Medical Research Council (MC_UU12018/7).

## Additional information

### Funding

| Funder | Grant reference number | Author |
|---|---|---|
| Cancer Research UK | FC001076 | Daniel Fisch<br>Barbara Clough<br>Joseph Wright<br>Monique Bunyan<br>Michael Howell<br>Eva Frickel |
| Wellcome Trust | FC001076 | Daniel Fisch<br>Barbara Clough<br>Joseph Wright<br>Monique Bunyan<br>Michael Howell<br>Eva Frickel |
| UK Medical Research Council | FC001076 | Daniel Fisch<br>Barbara Clough<br>Joseph Wright<br>Monique Bunyan<br>Michael Howell<br>Eva Frickel |
| Wellcome Trust | 091664/B/10/Z | Eva Frickel |
| Boehringer Ingelheim Fonds | | Daniel Fisch |
| European Research Council | 649101-UbiProPox | Jason Mercer |
| Medical Research Council | MC_UU12018/7 | Artur Yakimovich<br>Jason Mercer |

The funders had no role in study design, data collection and interpretation, or the decision to submit the work for publication.

## Author contributions
Daniel Fisch, Conceptualization, Data curation, Software, Formal analysis, Validation, Investigation, Visualization, Methodology, Writing—original draft, Writing—review and editing; Artur Yakimovich, Data curation, Software, Formal analysis, Validation, Methodology, Writing—original draft, Writing—review and editing; Barbara Clough, Joseph Wright, Monique Bunyan, Investigation, Writing—review and editing; Michael Howell, Resources, Methodology, Writing—review and editing, Provided essential high content imaging guidance and performed automated image acquisition; Jason Mercer, Funding acquisition, Writing—original draft, Writing—review and editing; Eva Frickel, Conceptualization, Data curation, Supervision, Funding acquisition, Writing—original draft, Writing—review and editing

## Author ORCIDs
Daniel Fisch http://orcid.org/0000-0002-8155-0367
Artur Yakimovich http://orcid.org/0000-0003-2458-4904
Barbara Clough http://orcid.org/0000-0002-3235-6170
Jason Mercer http://orcid.org/0000-0003-1466-9541
Eva Frickel http://orcid.org/0000-0002-9515-3442

## Decision letter and Author response
Decision letter https://doi.org/10.7554/eLife.40560.020
Author response https://doi.org/10.7554/eLife.40560.021

## Additional files
### Supplementary files
• Supplementary file 1. Overview and evaluation of existing software packages for analysis of fluorescence images in HCI experiments.
DOI: https://doi.org/10.7554/eLife.40560.015
• Transparent reporting form
DOI: https://doi.org/10.7554/eLife.40560.016

### Data availability
All data are contained within the manuscript. Due to the size of the dataset, the raw data are available upon request from the corresponding author but a 20Gb subset is available via Dryad (doi:10.5061/dryad.6vq2mp0).

The following dataset was generated:

| Author(s) | Year | Dataset title | Dataset URL | Database and Identifier |
|---|---|---|---|---|
| Fisch D, Yakimovich A, Clough B, Wright J, Bunyan M | 2019 | Data from: Defining host–pathogen interactions employing an artificial intelligence workflow | http://dx.doi.org/10.5061/dryad.6vq2mp0 | Dryad , 10.5061/dryad.6vq2mp0 |

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
