## [Decision Letter]

[**Editorial note:** This article has been through an editorial process in which the authors decide how to respond to the issues raised during peer review. The Reviewing Editor's assessment is that all the issues have been addressed.]

Acceptance notification:

The reviewers appreciate that you addressed all major concerns and have no experimental questions. The reviewers point out that your manuscript fits well as a Tools and Resources article, since it is a nice proof of concept study, but doesn't provide novel biological insights.

Decision letter after peer review:

Thank you for submitting your article "An Artificial Intelligence Workflow for Defining Host-Pathogen Interactions" for consideration by *eLife*. Your article has been reviewed by three peer reviewers, including Markus Meissner as the guest Reviewing Editor and Reviewer #1, and the evaluation has been overseen by a Detlef Weigel as the Senior Editor.

The Reviewing Editor has highlighted the concerns that require revision and/or responses, and we have included the separate reviews below for your consideration. If you have any questions, please do not hesitate to contact us.

Summary:

In their report Fisch et al., develop an open-source imaging platform (HRMAn) that integrates machine learning and deep learning, in order to analyse of multiple phenotypes related to host-pathogen interactions. The platform is openly available via KNIME Workflows, which will allow others to benefit the presented pipeline. The analysis follows basic principles, established by similar HCI platforms, involving illumination correction, followed by segmentation and infection detection (Stage 1). This is followed by an analysis of host protein recruitment, by detection of p62 at the parasitophorous vacuole. The authors also demonstrate that the analysis pipeline can be easily adjusted to other intracellular pathogens, such as *Salmonella* typhimurium.

Major concerns:

The reviewers raised some concerns regarding the technical novelty of this analysis pipeline. In particular analysis performed in Stage 1 is quite standard and follows similar methods as described before (see for example Touquet et al., 2018). The analysis in Stage 2, although well performed, does not represent a major methodogical advance in the field of machine learning.

Although the authors present some applications of this analysis pipeline, it would be advisable to provide additional data, demonstrating that HRMAn can detect additional, previously established phenotypes, such as those for ROP18 or ROP5.

Another application could be to probe for protein export from the parasite to the host cell, since this would broaden the analysis to detect additional host-pathogen interactions.

Given that HRMAn only provides a modest improvement over existing software, primarily in the user interphase, it would be appropriate to more extensively test its function in more authentic experimental settings (see also comments of reviewer 3).

The authors should also provide examples, why their analysis platform is superior. The authors acknowledge the existence of many other tools for high throughput analysis, but little to no discussion of similarities or difference to these tools is given.

While all reviewers agree that the experiments have been carefully performed, there is some scepticism regarding the novelty as a method (Reviewer 2) and the breadth of applications this tool will be used for in the field (Reviewer 1 and 3). As it stands, this tool appears to be optimised for the primary research interest of the authors, where it will certainly be very useful. However, it is unclear if other researchers in the field will recognise the advantage of HRMAn over existing HCI platforms (Reviewer 1 and 3).

Separate reviews (please respond to each point):

*Reviewer #1:*

Summary:

In their report Fisch et al., describe the establishment of a novel, open source programme for High Content Imaging (HCI). The major advantage of this system is the integration of machine learning, making it probably more adaptable to the analysis of different phenotypes related to host-pathogen interactions. The data and analysis appear to be of the highest quality and there is little doubt that HRMAn is a robust new tool to automatically detect phenotypes, which will be a valuable source for HCI. Especially since it is now possible to perform genome wide screens in *Toxoplasma gondii*, this pipeline comes in very handy for some researchers in order to perform phenotypic, image based screens on this parasite. The authors also demonstrate that the pipeline can be easily adapted to other pathogens, using *Salmonella* as a proof of principle.

Own opinion:

The described technology will be very useful for researchers planning to perform image based screens on host-pathogen interactions or on intracellular pathogens in general. Therefore, this study will be certainly of interest for a broad readership.

On the downside, it is a technological advance without novel biological insight. As the authors mention, there are many open source platforms available (not to mention commercial software) that allow HCI analysis of parasite growth, invasion, etc. (CellProfiler, CellClassifier, Fijii, etc.). A recent publication used a relatively simple HCI analysis to perform chemical screens in Toxoplsma gondii (see Touquet et al., 2018), which is certainly inferior to the platform presented in this study.

I am a bit unsure, how and why the provided pipeline is superior to other pipelines. At least the basic principles of HCI analysis appear similar:

– Pre-processing for illumination correction

– Segmentation (in this case host cell vs parasites)

– Analysis of pre-defined features (i.e. size of parasitophorous vacuoles, host cell nuclei, etc.)

As such I am not fully convinced if the platform is indeed superior to other imaging analysis software. For example in our lab we used Cell Profiler, which allows us to determine invasion rate, parasite growth rate, host cell number, etc. The authors should provide some examples or an in depth discussion regarding the advantages of their pipeline, when compared to other

Saying that, the described image analysis pipeline is very well designed and if widely used in the field will allow to analyse quantitative phenotypic data that are comparable in between different laboratories.

The manuscript is well written and the techniques used are of the highest quality. The presented data are very solid, demonstrating that this analysis pipeline is very accurate. Unfortunately I cannot comment on machine learning, since this is outside my area of expertise.

At this point the analysis pipeline is well suited for the analysis of host-pathogen interactions, in particular the characterisation of host-protein recruitment to the PV, a key interest of the Frickel lab and this aspect is of somewhat narrow interest.

It would be good to summarise in an additional figure, which phenotypes this imaging platform can differentiate.

For example, instead of host protein recruitment, it would be of great interest to also analyse parasite protein export into the host cell. Parasite lines expressing for example dense granule proteins are well described in the literature and it should be straight forward to add this parameter to the analysis pipeline.

In summary, it is a well described analysis pipeline for HCI that might find broad applications, especially now, that genome wide screens can be performed using CRISPR/cas9.

*Reviewer #2:*

The proposed analysis will be made openly available via KNIME Workflows. This is certainly a plus since it will allow others to benefit from the presented analysis pipeline. Additionally, KNIME is easy to install, with a bit of practice very intuitive, and could on demand also be used to change/extend the workflow.

The overall analysis pipeline can be divided in 3 steps. An illumination correction step followed by a segmentation and infection detection step (named 'Stage 1'), followed up by an analysis of host protein recruitment ('Stage 2').

The analysis in 'Stage 1' is not bearing surprises or any methodic novelty. The proposed pipeline, a combination of default analysis components, solves the task at hand as long as the provided images (after illumination correction) can be segmented via a simple threshold. The merit of 'Stage 1', as mentioned above, is clearly not any methodological advance or interesting combination of existing methods. It is, never the less, sensible engineering work and might very well help others that desire to perform the same analysis.

In case a host protein recruitment analysis is desired, a 'Stage 2' workflow is proposed. Besides a decision tree that is used in Stage 1, this is the only place where any machine learning technique is applied. More specifically, a variant of the well known AlexNet is used to learn to classify protein recruitment. The relatively shallow network architecture and all parameter and training decisions are sensible set to values used in many neural network applications. While, as before, nothing here is even close to being a methodological advance in the field of machine learning, all decisions seem well thought trough and I have no problems believing that final classification results are good.

It might be desirable to compare the proposed analyses system (or its components) to other existing systems or modules. (The authors acknowledges the existence of many other tools for high throughput analysis, see Supplementary file 1, but little to no discussion of similarities or difference to these tools is given.)

The impact of this work will depend on the presented data and the utility of the proposed analysis pipeline (which is/will be openly available). I am, unfortunately, not the right person to judge how many research project are currently just waiting to use an analysis pipeline as the one presented in this manuscript.

I would like to end my review by stating my biggest concern. After reading the title and Abstract of this manuscript, I expected the presented work to be significantly more involved. It turns out that the presented workflow uses one decision tree in Stage 1 and an 'off the shelve' AlexNet in the optional Stage 2. I would advice the authors to tone down the paper pitch in this regard.

In summary, I do not believe that the presented manuscript contains enough methodological advances to justify publication of the method alone. If the presented data and performed analyses justifies publication will have to be judged by a person in the field of image-based infection biology.

Minor Comments:

The last sentence in the Abstract makes a bold statement about 'operating at human capability'. This was in my point of view not shown in the manuscript and therefore ends up being a bold claim lacking justification.

While I could follow all explanations about network training, some formulations could benefit from feedback by a person with publishing experience in the field of machine learning.

*Reviewer #3:*

The authors develop an open-source image analysis platform named HRMAn that relies on machine learning algorithms and deep learning. Given input images, the platform characterizes phenotypes such as parasites/vacuole, vacuole size, and host-protein recruitment. The platform is also high-throughput, allowing for the bulk submission of many images simultaneously. To validate the platform, the authors trained the algorithm with an annotated dataset of host cells infected by eGFP-expressing *Toxoplasma gondii* parasites. Using this training dataset, the authors quantified differences in parasite killing/growth restriction strategies across host cell lines and between Type I and Type II Toxoplasma lines. The authors also validated the algorithm's ability to quantify levels of ubiquitin and p62 recruitment to the PVM. Finally, the authors retrained the algorithm using an annotated dataset of *Salmonella* typhimurium, demonstrating the ability of HRMAn to recognize and quantify a diversity of pathogens.

HRMAn provides a high-throughput and effective strategy for analyzing phenotypes by microscopy. The platform removes the human component of analysis and bulk-input allows for the rapid analysis of thousands of cells. The interface is remarkably easy to set-up and navigate, particularly alongside the available tutorials. The demonstration of success with two pathogens of drastically different size, *Toxoplasma gondii* and *Salmonella* typhimurium, suggests that this could be a valuable tool for a wide array of microbes. While the recruitment of host-proteins to the PVM is a potentially powerful tool, it is currently limited to binary processes (presence or absence of ubiquitin/p62). As shown, it is unclear if HRMAn can detect more subtle, non-binary phenotypes, such as recruitment of mitochondria. The authors demonstrate the ability of HRMAn to quantify differences in ubiquitin and p62 recruitment between both different host cell and different parasite lines upon IFNγ-priming. It would be desirable to demonstrate HRMAn's ability to detect previously established phenotypes, such as those of ROP18 or ROP5. This will more strongly validate HRMAn's potential ability to detect novel phenotypes. Given that HRMAn only provides a modest improvement over existing software, primarily in the user interphase, it would be appropriate to more extensively test its function in more authentic experimental settings.

Major concerns:

a) Since there are many differences between Type I and Type II parasites (including growth rate and viability, which could affect the recruitment measurements), it would be appropriate for the authors to look at isogenic lines that differ only in a particular effector. The authors could test HRMAn's ability to detect previously known phenotypes, such as the increase in GBP recruitment in a ROP18 knockout.

b) It is unclear why the authors, when classifying parasites/vacuole, bin the vacuoles by 1, 2, 4, and greater than 4. The authors should provide rationale, technical or otherwise, for binning all vacuoles >4 together, since some phenotypes might emerge only later during intracellular growth.

c) In Figure 2C, the example picture for >4/vac appears to be a picture of a vacuole containing 4 parasites. This should be corrected.

d) It is important that the training dataset used for the manuscript be released in its entirety to ensure that readers can replicate the results of the paper and account for any differences between lab-specific assignments and HRMAn.

Minor concerns:

a) The authors often use the phrase "cell" ambiguously and it is unclear if they are referring to host cells or parasite cells. The authors should take care to reduce ambiguity by more clearly stating which cells they are referring to.

b) In Figure 2E, the heat map of the confusion matrix is difficult to accurately assess due to the similarity between many values. Number values should be provided as well, or in place of the matrix.

c) In Figure 2—figure supplement 1A, images displaying separation of the channels should be provided. As such, it is difficult to evaluate the presence of GRA2. The current image does not seem to accurately represent the 98% prevalence of GRA2+ vacuoles. By separating the image channels this should be more apparent.

---

## [Author Response]

Major concerns:The reviewers raised some concerns regarding the technical novelty of this analysis pipeline. In particular analysis performed in Stage 1 is quite standard and follows similar methods as described before (see for example Touquet et al., 2018).

Thank you very much for your comments and your in-depth review of our manuscript. We agree that stage 1 of the analysis uses well-established image analysis methods. Stage 1 of the image analysis is a necessity for stage 2, hence, it is presented in the manuscript. It is an easy-to-use image analysis pipeline useful for host-pathogen researchers wanting to screen larger imaging datasets, e.g. siRNA or CRISPR screens. We therefore offer the option of just running this part of the analysis for the users’ convenience.

The analysis in Stage 2, although well performed, does not represent a major methodogical advance in the field of machine learning.

Yes, that is correct, stage 2 of HRMAn does not offer a stand-alone major methodological advance in the field of machine learning. The goal of stage 2 HRMAn is to make accessible and tailor a neural network-driven analysis to the researcher to define host-pathogen interaction in an unbiased, highly reproducible, sound and robust fashion. A neural network-driven pipeline is the only way to do this in an automated fashion and this approach has not been employed for the large field of host-pathogen interaction before.

Having said that, the deep neural network used in this work is based on a published AlexNet architecture, yet we have changed and adopted it for the problem setting (see Figure 1). More precisely, we have added a detailed description of the derivation of HRMAlexNet in the Materials and methods section under “Deep Learning Setup and Neural Network Architecture”. Instead of optimising our architecture using off-theshelf open datasets we worked with real world data. Hence both problem and solution presented are novel. The adaptation of the network and integration into a user-friendly program makes it suitable for non-specialists and the wider research community. This was the main objective from the outset of this work – rather than advance AI, we wanted to tailor it to host-pathogen questions and make it available to biologists.

We understand however, how semantics could potentially lead to confusion within the field of machine learning. To denote that we indeed have tailored a neural network approach to a problem rather than newly developed from scratch a neural network-driven solution, we have changed the title to: “Defining Host-Pathogen Interactions Employing an Artificial Intelligence Workflow”.

Although the authors present some applications of this analysis pipeline, it would be advisable to provide additional data, demonstrating that HRMAn can detect additional, previously established phenotypes, such as those for ROP18 or ROP5.

We have added additional data to illustrate that HRMAn can indeed detect previously established phenotypes. As suggested by the reviewer, we specifically addressed the role of *Tg* virulence proteins within murine host defence. We repeated an experiment from Virreira Winter et al., 2011 demonstrating the influence of polymorphic, secreted *Tg* effector proteins ROP16, ROP18 and GRA15 on the recruitment of mGbps to the *Tg* vacuole. We obtained comparable results as previously published and extended the previously published results to two additional mGbps using unbiased HRMAn analysis. This data is now presented in Figure 5A.

Another application could be to probe for protein export from the parasite to the host cell, since this would broaden the analysis to detect additional host-pathogen interactions.

We thank the reviewers for this point and agree this would nicely expand our analysis pipeline to include parasite-derived factors. In the previously submitted version our work exclusively looked at the host cell response. We therefore performed an analysis of *Tg* effector protein secretion and translocation to the nucleus. To do so, we infected HFF cells using type I *Tg* (RH) that express HA-tagged GRA16, GRA24 and type II *Tg* (Pru) that express HA-tagged *Tg*IST. Using HRMAn, we readily determined the intracellular distribution of the effector proteins and studied the relationship between uninfected and infected cells. Furthermore, HRMAn correlated the amount of effector protein fluorescence signal to the number of parasites in the cell. This data is now presented in Figure 5B.

Given that HRMAn only provides a modest improvement over existing software, primarily in the user interphase, it would be appropriate to more extensively test its function in more authentic experimental settings (see also comments of reviewer 3).

Here we beg to differ with the reviewers. Yes, HRMAn does not develop novel machine learning algorithms. However, it presents a beyond state-of-the-art image analysis pipeline for host-pathogen interaction. HRMAn, for the first time, brings AI to image analysis of host-pathogen interaction and does so in a user-friendly interface making it accessible to researchers without the ability to code. This is a major advance and advantage over currently available programs.

However, we do agree with the reviewers that to further test HRMAn’s function in what reviewer 3 calls more authentic experimental settings is of benefit for the manuscript and the use of the program. We therefore added additional sets of analysis as described above (Figure 5A and 5B) specifically comparing isogenic *Toxoplasma* strains and we have added a murine cell line to the analysis. Collectively the analysis presented now encompasses a total of nine *Tg* strains, *Salmonella enterica* Typhimurium and 6 different cell lines from two species.

The authors should also provide examples, why their analysis platform is superior. The authors acknowledge the existence of many other tools for high throughput analysis, but little to no discussion of similarities or difference to these tools is given.

We agree with the reviewers. To complement Supplementary file 1, we have added further descriptions of these programs in the Introduction. We have also added a comparison of our analysis platform versus what is currently available in the Introduction: “Solutions existing to date can largely be split into two major categories: user-friendly turn-key GUI (TK-GUI)-based solutions and scripts ensembles (SE) solutions. Due to the large support burden of the TK-GUI, these programs generally lack the implementations of the latest engineering advances.”

While all reviewers agree that the experiments have been carefully performed, there is some scepticism regarding the novelty as a method (Reviewer 2) and the breadth of applications this tool will be used for in the field (Reviewer 1 and 3). As it stands, this tool appears to be optimised for the primary research interest of the authors, where it will certainly be very useful. However, it is unclear if other researchers in the field will recognise the advantage of HRMAn over existing HCI platforms (Reviewer 1 and 3).

We thank the reviewers for their careful assessments. We think we have addressed the comments on the novelty of the method, as well as expanded its breadth of applications (see all points above). Given that HRMAn is the only program that provides the capability of analysing protein targeting by neural networks it is more sophisticated than existing image analysis solutions. In support of its more than 1,200 views and 360 full pdf downloads on BioRxiv in 8 weeks (22.11.2018), suggests that HRMAn is already being employed by the field. In addition, besides *Toxoplasma* and *Salmonella*, we are already expanding HRMAn for analysis of other pathogens, including *Chlamydia* and *Cryptococcus* in collaboration with other labs interested in using HRMAn for their experiments.

We would like to take the opportunity here to underscore that host protein recruitment to pathogens is an expanding field. While the reviewers did not comment on the fact that HRMAn can be employed for analysis of *Salmonella*, the anti-*Salmonella* immunity field is largely focused on vacuole recruitment and marking of cytosolic bacteria. Being able to recognise two pathogens with a magnitude of difference in size points to the fact that HRMAn can be adapted to an ever expanding range of pathogens, making it an attractive and important advance for the host-pathogen community at large.

Separate reviews (please respond to each point):

Reviewer #1:

[…] Own opinion:The described technology will be very useful for researchers planning to perform image based screens on host-pathogen interactions or on intracellular pathogens in general. Therefore, this study will be certainly of interest for a broad readership.

We thank the reviewer for their assessment and concur that HRMAn is useful for both *Toxoplasma* and *Salmonella* researchers.

On the downside, it is a technological advance without novel biological insight.

The focus of our manuscript was placed on the development and optimization of HRMAn as an image analysis pipeline. Having said that, there is in fact novel biological insight in this study. This is the first time 5 different human cell lines have been tested side-by-side for their response to *Toxoplasma*, and titrated for the IFNgdependent anti-*Toxoplasma* response. This data alone is of great use to the *Toxoplasma* immunity field as we demonstrate that various cell lines require different levels of IFNγ to mount an efficient anti-*Toxoplasma* response.

Further, by carefully assessing a larger dataset of ubiquitin recruitment to cytosolic *Salmonella* in HeLa cells than has ever been analysed before, we find that HeLa cells seem to have reached their capacity of clearing *Salmonella* by ubiquitination and subsequent autophagy, independent of IFNγ treatment. This is a finding that has been missed by other publications before, due to insufficient numbers of bacteria manually counted.

As the authors mention, there are many open source platforms available (not to mention commercial software) that allow HCI analysis of parasite growth, invasion, etc. (CellProfiler, CellClassifier, Fijii, etc.). A recent publication used a relatively simple HCI analysis to perform chemical screens in Toxoplsma gondii (see Touquet et al., 2018), which is certainly inferior to the platform presented in this study.I am a bit unsure, how and why the provided pipeline is superior to other pipelines. At least the basic principles of HCI analysis appear similar:– Pre-processing for illumination correction– Segmentation (in this case host cell vs parasites)– Analysis of pre-defined features (i.e. size of parasitophorous vacuoles, host cell nuclei, etc.)As such I am not fully convinced if the platform is indeed superior to other imaging analysis software. For example in our lab we used Cell Profiler, which allows us to determine invasion rate, parasite growth rate, host cell number, etc. The authors should provide some examples or an in depth discussion regarding the advantages of their pipeline, when compared to other

As detailed above, stage 1 uses well-established image analysis methods. As it is a necessity for stage 2, it is presented in the manuscript. For simpler experiments it is an easy-to-use tool useful for host-pathogen researchers screening larger imaging datasets, e.g. siRNA or CRISPR screens. We therefore offer the option of running this part of the analysis for the users’ convenience.

Stage 2 of the analysis pipeline – the stage that assesses the host-parasite interaction dimension – is the innovation of HRMAn. Rather than simple pixel enumeration, stage 2 employs a deep learning neural network to assess host protein recruitment to the pathogen. This is currently not achievable by any image analysis program available, neither commercial nor open-source.

As said above, we agree with the reviewer and to complement Supplementary file 1, we have added further descriptions of these programs in the Introduction. We have also added a comparison of our analysis platform versus what is currently available in the Introduction: “Solutions existing to date can largely be split into two major categories: user-friendly turn-key GUI (TK-GUI)-based solutions and scripts ensembles (SE) solutions. Due to the large support burden of the TK-GUI, these programs generally lack the implementations of the latest engineering advances.”

Saying that, the described image analysis pipeline is very well designed and if widely used in the field will allow to analyse quantitative phenotypic data that are comparable in between different laboratories.The manuscript is well written and the techniques used are of the highest quality. The presented data are very solid, demonstrating that this analysis pipeline is very accurate. Unfortunately I cannot comment on machine learning, since this is outside my area of expertise.At this point the analysis pipeline is well suited for the analysis of host-pathogen interactions, in particular the characterisation of host-protein recruitment to the PV, a key interest of the Frickel lab and this aspect is of somewhat narrow interest.It would be good to summarise in an additional figure, which phenotypes this imaging platform can differentiate.

Thank you for your comment and thought. In fact, all readouts that HRMAn produces during the image analysis are already summarised in Figure 1.

We have added another figure of additional data showing how HRMAn can easily be used to study pathogen biology as well (Figure 5). The new experiments for this figure are described above. Many of the readouts that HRMAn calculates can be used for both parameters of an infection, e.g. the readout of recruitment of host proteins to the vacuole can equally be used to study display or absence of pathogen proteins at the vacuole. This only requires different staining of the microscopy samples. The same can be said about the replication and growth characterisation of pathogens. The measurements performed by HRMAn could be used to study the effect of pathogen gene knockouts or protein depletion that result in growth phenotypes.

We want to bring to the reviewer’s attention, as detailed above, that HRMAn not only works for analysis of *Toxoplasma* infection, but also *Salmonella*. Cumulatively, these two fields account for a large community of researchers that share an interest in host defence to pathogens. As we demonstrate the versatility of HRMAn in this manuscript by choosing two pathogens of very different size, it is easy to envision that HRMAn will be adaptable to other pathogens in the future.

For example, instead of host protein recruitment, it would be of great interest to also analyse parasite protein export into the host cell. Parasite lines expressing for example dense granule proteins are well described in the literature and it should be straight forward to add this parameter to the analysis pipeline.

As detailed above, we now demonstrate that HRMAn can analyse the localisation of *Toxoplasma* proteins in host cells including GRA16, GRA24 and *Tg*IST. This is now included as Figure 5A.

In summary, it is a well described analysis pipeline for HCI that might find broad applications, especially now, that genome wide screens can be performed using CRISPR/cas9.

We thank the reviewer for his assessment.

Reviewer #2:

The proposed analysis will be made openly available via KNIME Workflows. This is certainly a plus since it will allow others to benefit from the presented analysis pipeline. Additionally, KNIME is easy to install, with a bit of practice very intuitive, and could on demand also be used to change/extend the workflow.The overall analysis pipeline can be divided in 3 steps. An illumination correction step followed by a segmentation and infection detection step (named 'Stage 1'), followed up by an analysis of host protein recruitment ('Stage 2').The analysis in 'Stage 1' is not bearing surprises or any methodic novelty. The proposed pipeline, a combination of default analysis components, solves the task at hand as long as the provided images (after illumination correction) can be segmented via a simple threshold. The merit of 'Stage 1', as mentioned above, is clearly not any methodological advance or interesting combination of existing methods. It is, never the less, sensible engineering work and might very well help others that desire to perform the same analysis.

As mentioned above, stage 1 of the analysis uses well-established image analysis methods and was not touted as novel methodology. Stage 1 is essential for stage 2, with the advantage that it provides an easy-to-use, robust tool for screening large imaging datasets.

In case a host protein recruitment analysis is desired, a 'Stage 2' workflow is proposed. Besides a decision tree that is used in Stage 1, this is the only place where any machine learning technique is applied. More specifically, a variant of the well known AlexNet is used to learn to classify protein recruitment. The relatively shallow network architecture and all parameter and training decisions are sensible set to values used in many neural network applications. While, as before, nothing here is even close to being a methodological advance in the field of machine learning, all decisions seem well thought trough and I have no problems believing that final classification results are good.

Thank you for your comment. We would like to emphasise that HRMAn has been designed for biologist. As such, we decided to keep the machine learning algorithms as simple as possible, to save on computing resources and to prevent overcomplicating the analysis and its set-up. As shown for the decision tree classifier, we tested different commonly used algorithms and settings and chose the simplest and most robust option.

It is worth mentioning, that the neural network architecture presented in the current manuscript is different from the original AlexNet and adopted for the problem setting (see Figure 1). While having the same number of convolutional layers, their dimensions are different. Furthermore, unlike the network presented by Krizhevsky et al., it is not parallelised in two streams aimed for two GPUs. Instead the optimisation of our network runs on a single GPU making it more accessible for the end user (setups with dual GPUs are quite rare and difficult to operate in biological labs). It mimics the AlexNet in an effort to minimize the number of hyperparameters to optimize. Given the number of hyperparameter changes it is a new architecture. Provided the fact that the network has been optimized on the real-world data, rather than publicly available datasets, proved to be solvable, alighting as close as possible to a known solution is dictated by the good engineering practice. Nonetheless beyond architectural novelty, HRMAn is a framework allowing for swappable architectural solutions. It supports state-of-the-art libraries like DL4J. Keras is upcoming, which will allow rapid adoption of most advance algorithms on the realworld data.

We have now detailed the exact modifications we performed to create HRMAlexNet – the neural network architecture of HRMAn in the Materials and methods section under the points “Deep Learning Setup and Neural Network Architecture” and “Neural Network Training and Hyperparameters Optimization”.

It might be desirable to compare the proposed analyses system (or its components) to other existing systems or modules. (The authors acknowledges the existence of many other tools for high throughput analysis, see Supplementary file 1, but little to no discussion of similarities or difference to these tools is given.)

As suggested, to complement Supplementary file 1, we now review other analysis solutions in light of HRMAn in the Introduction part as mentioned above.

The impact of this work will depend on the presented data and the utility of the proposed analysis pipeline (which is/will be openly available). I am, unfortunately, not the right person to judge how many research project are currently just waiting to use an analysis pipeline as the one presented in this manuscript.

Since its deposition on BioRxiv the manuscript has been downloaded 360 and the Abstract read more than 1200 times (22.11.2018). We have already received more than 10 requests for early-access to the GitHub repository. HRMAn is currently being used and further developed in 7 labs, that we know of, prior to official publication (22.11.18).

I would like to end my review by stating my biggest concern. After reading the title and Abstract of this manuscript, I expected the presented work to be significantly more involved. It turns out that the presented workflow uses one decision tree in Stage 1 and an 'off the shelve' AlexNet in the optional Stage 2. I would advice the authors to tone down the paper pitch in this regard.

As pointed out above, to not mislead the reader, we have re-titled our manuscript “Defining Host-Pathogen Interactions Employing an Artificial Intelligence Workflow”.

We would like to point out that beyond designing HRMAlexNet, as mentioned above we have presented a novel methodology of visualization of the phenotypic attention of the trained neural network. Through class-activation map generation (occlusion map method) we show the fraction of the pathogen micrograph important for evaluation of the host-pathogen interaction (Figure 2). In our opinion, this pushes the boundary of the conventional machine learning toward artificial intelligence-based analysis. Unbiased attention visualization allows us to leverage on the unsupervised component of the HRMAlexNet responsible for learning from the presented data to promote data-driven discovery. We have now amended the text to point out these advances/differences.

In summary, I do not believe that the presented manuscript contains enough methodological advances to justify publication of the method alone. If the presented data and performed analyses justifies publication will have to be judged by a person in the field of image-based infection biology.

Considering the reviewers concerns, we have significantly improved the presentation of HRMAn through additional experimental analysis, a refined description of HRMAn and its novelty, as well as by comparing our analysis solution to other existing programs.

Minor Comments:The last sentence in the Abstract makes a bold statement about 'operating at human capability'. This was in my point of view not shown in the manuscript and therefore ends up being a bold claim lacking justification.

Thank you for your comment, but we feel this statement is justified. We demonstrate in our cross-validation 92% accuracy with the use of trained CNNs. This high accuracy combined with the throughput of HRMAn (classification of more than 100,000 vacuoles in just one afternoon) in fact far exceeds human capabilities.

While I could follow all explanations about network training, some formulations could benefit from feedback by a person with publishing experience in the field of machine learning.

The manuscript has now been revised by a Google engineer specializing in CNNs and updated accordingly.

Reviewer #3:

[…] Major concerns:a) Since there are many differences between Type I and Type II parasites (including growth rate and viability, which could affect the recruitment measurements), it would be appropriate for the authors to look at isogenic lines that differ only in a particular effector. The authors could test HRMAn's ability to detect previously known phenotypes, such as the increase in GBP recruitment in a ROP18 knockout.

As detailed above, we have repeated the experiment of mGbps recruitment to the Tg vacuole presented in Virreira Winter et al., 2011. The influence of polymorphic, secreted *Tg* effector proteins ROP16, ROP18 and GRA15 on the mGbp1 recruitment phenotype was demonstrated in this publication. We find that HRMAn produces identical results as manual scoring by humans, however, the number of vacuoles analysed far exceeds what has been done previously. Additionally, we now analysed not only mGbp1, but also mGbp2 and 5 recruitment to the PV. The new data is now presented in Figure 5A.

b) It is unclear why the authors, when classifying parasites/vacuole, bin the vacuoles by 1, 2, 4, and greater than 4. The authors should provide rationale, technical or otherwise, for binning all vacuoles >4 together, since some phenotypes might emerge only later during intracellular growth.

Vacuoles were binned into 4 or more parasites as when vacuoles contain more than 4 parasites it is difficult to distinguish, even by eye, exactly how many parasites are contained within. For our purposes, this would result in unreliable data. That said, use of KNIME Analytics platform will allow users to easily adjust and re-train their classifiers, according to their data confidence level.

c) In Figure 2C, the example picture for >4/vac appears to be a picture of a vacuole containing 4 parasites. This should be corrected.

Thank you for the comment, we have replaced the image with an image that has more than four parasites in the same vacuole and added scale bars to all images.

d) It is important that the training dataset used for the manuscript be released in its entirety to ensure that readers can replicate the results of the paper and account for any differences between lab-specific assignments and HRMAn.

The trained CNNs will be distributed through our own homepage hrman.org and the GitHub repository. Additionally, we have now stated that “images used for training of the machine learning algorithms will be available through the Crick Institute’s online file exchange system upon request.” in the Materials and methods section.

Minor concerns:a) The authors often use the phrase "cell" ambiguously and it is unclear if they are referring to host cells or parasite cells. The authors should take care to reduce ambiguity by more clearly stating which cells they are referring to.

Thank you for your comment. We have updated the text to more clearly distinguish between the host cell and pathogens.

b) In Figure 2E, the heat map of the confusion matrix is difficult to accurately assess due to the similarity between many values. Number values should be provided as well, or in place of the matrix.

We thank the reviewer for this comment and have now added values into the panels of Figure 2E and Figure 4D to make it easier for the reader to assess the accuracy of the classification.

c) In Figure 2—figure supplement 1A, images displaying separation of the channels should be provided. As such, it is difficult to evaluate the presence of GRA2. The current image does not seem to accurately represent the 98% prevalence of GRA2+ vacuoles. By separating the image channels this should be more apparent.

Thank you for the suggestion. We have added the individual channels as black and white images for easier visualisation and inspection by the reader.